# Motivational signals disrupt metacognitive signals in the human ventromedial prefrontal cortex

Monja Hoven [1✉], Gina Brunner[1,2], Nina S. de Boer[1,3], Anna E. Goudriaan[1,4], Damiaan Denys[1,5], Ruth J. van Holst[1,8], Judy Luigjes[1,8] & Maël Lebreton[6,7,8]

A growing body of evidence suggests that, during decision-making, BOLD signal in the ventromedial prefrontal cortex (VMPFC) correlates both with motivational variables – such as incentives and expected values – and metacognitive variables – such as confidence judgments – which reflect the subjective probability of being correct. At the behavioral level, we recently demonstrated that the value of monetary stakes bias confidence judgments, with gain (respectively loss) prospects increasing (respectively decreasing) confidence judgments, even for similar levels of difficulty and performance. If and how this value-confidence interaction is reflected in the VMPFC remains unknown. Here, we used an incentivized perceptual decision-making fMRI task that dissociates key decision-making variables, thereby allowing to test several hypotheses about the role of the VMPFC in the value-confidence interaction. While our initial analyses seemingly indicate that the VMPFC combines incentives and confidence to form an expected value signal, we falsified this conclusion with a meticulous dissection of qualitative activation patterns. Rather, our results show that strong VMPFC confidence signals observed in trials with gain prospects are disrupted in trials with no – or negative (loss) – monetary prospects. Deciphering how decision variables are represented and interact at finer scales seems necessary to better understand biased (meta) cognition.

[1] Department of Psychiatry, Amsterdam UMC, University of Amsterdam, Amsterdam, The Netherlands. [2] Institute of Neuroscience and Psychology, University of Glasgow, Glasgow, UK. [3] Department of Philosophy, Radboud University, Nijmegen, The Netherlands. [4] Arkin and Jellinek, Mental Health Care, Amsterdam, The Netherlands. [5] Netherlands Institute for Neuroscience, an Institute of the Royal Netherlands Academy of Arts and Sciences, Amsterdam, The Netherlands. [6] Swiss Center for Affective Science, University of Geneva, Geneva, Switzerland. [7] Laboratory for Behavioral Neurology and Imaging of Cognition, Department of Fundamental Neurosciences, University of Geneva, Geneva, Switzerland. [8] These authors contributed equally: Ruth van Holst, Judy Luigjes, Maël Lebreton. ✉email: m.hoven@amsterdamumc.nl

Over the past decades, a growing number of neurophysiological studies in human and non-human primates have established that the neural signals recorded during learning and decision-making tasks in the orbito-medial parts of the prefrontal cortex (OMPFC)—the medial orbitofrontal cortex (OFC) and the ventromedial prefrontal cortex (VMPFC)—correlate with key concepts from theories of motivation and decision-making[1–3]. For instance, in Pavlovian conditioning tasks, the activity of neurons in the non-human primate OFC correlates with the anticipatory value of upcoming rewards, with neural activity predicting the monkeys' subjective preferences[4]. In economic decision-making tasks, neuronal activity in the same region of the OFC correlates with the subjective value of available options[5]. In humans, similar results have been derived from functional neuroimaging studies. Blood-oxygen level-dependent (BOLD) signal in the VMPFC scales with the anticipation of upcoming rewards[6,7], the subjective pleasantness and desirability attributed to different stimuli[8], the willingness to pay for different types of goods[9–11], and the expected value (EV) of prizes, performance incentives, and economic bundles such as lotteries[12–15]. Overall, together with the midbrain and the ventral striatum (VS), the VMPFC seems to form a "brain valuation system"[16–18], whose activity automatically indexes the value of available options so as to guide value-based decision-making[8,10] and motivate motor and cognitive performance[19].

Recently, a set of human neurophysiological studies have suggested that activity in the VMPFC is also related to metacognitive processes[20,21]. In particular, both single neuron and BOLD activity in the VMPFC correlate with participants' confidence in their own judgments and choices[22–25]. Confidence is a metacognitive variable that can be defined as one's subjective estimate of the probability of a given choice being correct[26,27]. Just like values, confidence judgments seem to be automatically represented in the VMPFC, for different types of judgments and choices[24,28,29]. Confidence signals could be useful for the flexible adjustment of behavior—such as monitoring and reevaluating previous decisions[30], tracking changes in the environment[31,32], adapting future decisions[30,33], or arbitrating between different strategies[34,35].

Interestingly, at the behavioral level, values and confidence seem to interact. For instance, a handful of studies in psychology and economics have documented that positive incentive values, operationalized as prospects of monetary bonuses, increase subjective estimates of confidence[36]. Similar confidence boosts have been reported with higher state values, operationalized as positive incidental psychological states such as elevated mood[37], absence of worry[38], and emotional arousal[39–41]. Recently, we designed an incentivized perceptual decision-making task to demonstrate that monetary incentives bias confidence judgments, with gain (respective loss) prospects increasing (respectively decreasing) confidence judgments, even for similar levels of difficulty and performance[42]. This result was also replicated in a reinforcement-learning context[43,44]. We explicitly hypothesized that this interaction would stem from the concurrent neural representation of—hence putative interaction between—incentive values and confidence in the VMPFC[42].

Here, we used a functional neuroimaging adaptation of our original perceptual decision-making paradigm that allows for investigation of the overlap in neural correlates between incentive value and confidence[42]. Our first set of analyses did not show the hypothesized overlap of incentive value and confidence signals in the VMPFC at the expected statistical threshold ($p < 0.05$ whole-brain corrected family-wise error (FWE) at the cluster level), nor in other regions of interest (ROI) that have been linked with value, motivation, and confidence in the past—such as the VS and the anterior cingulate cortex (ACC). Therefore, we formulated an alternative hypothesis, positing that VMPFC integrates confidence and incentive signals into a probabilistic EV signal. We ran several quantitative and qualitative analyses that thoroughly compared the relative merits of these different hypotheses for the neural basis of the value-confidence interaction. Our results ultimately depict a complex picture, suggesting that motivational signals (notably prospects of loss) can disrupt metacognitive signals in the VMPFC.

## Results

To investigate the neurobiological basis of the interactions between incentives and confidence, we modified the task used in Lebreton et al.[42] to make it suitable for functional magnetic resonance imaging (fMRI) (Fig. 1a). Basically, this task is a simple perceptual task (contrast discrimination), featuring a two-alternative forced-choice followed by a confidence judgment. Then, we experimentally manipulated the available monetary outcomes, defining several incentive conditions: at each trial, participants could win (gain context) or lose (loss context) points —or not gain or lose anything (neutral context)—depending on the correctness of their choice. Incentives were presented in an interleaved fashion, in order to avoid contextualization of outcomes (rather than in a blocked design, where the absence of gain could be reframed as relative loss in a gain block, or vice versa). Importantly, this incentivization was implemented after the moment of choice and before the confidence rating. Consequently, by design, there should not be any incentivization effects on either accuracy or reaction times (RTs) as they develop during the choice. Note that this design corresponds to the simplest implementation of the task—corresponding to Experiment 2 in Lebreton et al.[42]— which otherwise conditioned monetary outcomes to confidence rating precision rather than choice accuracy (for details see ref. [42]). Yet, our previous results suggested that this task still reveals an effect of incentives on confidence, while keeping instructions simpler—a desirable feature, especially for clinical and fMRI studies.

**Behavioral results**. To start, we verified that our task generated the incentive-confidence interaction at the behavioral level. First, using an approach similar to Lebreton et al.[42], we used linear mixed-effects models to evaluate the effects of our experimental manipulation of incentives (i.e., the incentive condition) on behavioral variables (see Methods). More specifically, we defined and tested the incentives' biasing effects (i.e., the net incentive value, or in other words, the linear effect of incentives coded as −1, 0, and +1) and incentives' motivational effects (i.e., the absolute incentive value, or in other words, the mere presence of incentives, indicating whether something is at stake coded as 0 and +1). Replicating our previous results, we found a significant positive effect of incentive net value on confidence ($\beta = 0.78 \pm 0.32$, $t_{4317} = 2.43$, $p = 0.015$; Fig. 1b, c) and no effect of incentive absolute value ($\beta = -0.32 \pm 0.55$, $t_{4317} = -0.58$, $p = 0.565$; Fig. 1c). This result alone validates the presence of an incentive-confidence interaction at the behavioral level. Importantly, this effect was not driven by any net incentive value effects on accuracy or RT (accuracy: $\beta = 0.38 \pm 0.93$, $t_{4317} = 0.41$, $p = 0.685$; RT: $\beta = 13.75 \pm 19.22$, $t_{4317} = 0.72$, $p = 0.474$). Moreover, we did not find evidence for an effect of absolute incentive value on both accuracy and RT (accuracy: $\beta = 1.86 \pm 1.45$, $t_{4317} = 1.28$, $P = 0.199$; RT: $\beta = -25.24 \pm 29.17$, $t_{4317} = -0.87$, $p = 0.387$). Next, to confirm the robustness of our main effect of net incentive value on confidence, we ran several full linear mixed-effects models, which included additional control variables that could influence confidence as well (evidence, accuracy, RTs, et cetera, see Supplementary Note 1). Overall, the incentive-

**Fig. 1 Experimental design and behavioral results. a** Experimental paradigm. Participants viewed two Gabor patches on both sides of the screen (150 ms) and then chose which had the highest contrast (left/right, self-paced). After a jitter of a random interval between 4500 and 6000 ms, the incentive condition was shown (900 ms; green frame for win trials, gray frame for neutral trials, red frame for loss trials). Afterwards, participants were asked to report their confidence in the earlier made choice on a scale ranging from 50% to 100% with steps of 5%. The initial position of the cursor was randomized between 65% and 85%. Finally, subjects received feedback. The inter-trial interval (ITI) had a random duration between 4500 and 6000 ms. The calibration session only consisted of Gabor discrimination, without confidence rating, incentives, or feedback, and was used to adjust the difficulty so that every individual reached a performance of 70%. **b** Behavioral results. Individual-averaged accuracy (left), reaction times (middle) and confidence (right) as a function of incentive condition (−100/red, 0/gray, +100/green). Colored dots represent individuals ($N = 32$), gray lines highlight within-subject variation across conditions. Error bars represent sample mean ± standard error of the mean. Note that for confidence and accuracy, we computed the average per incentive level per individual, but that for reaction times, we computed the median for each incentive condition rather than the mean due to their skewed distribution. **c** Linear mixed-effect model (LMEM) results. The graph depicts fixed-effect regression coefficients ($\beta$) for incentive condition (Inc.) and absolute incentive condition (|Inc.|) predicting performance (top), reaction times (middle), and confidence (bottom). Error bars represent standard errors of fixed effects. *$p < 0.05$.

confidence interaction remained significant after accounting for those other potential sources of biases and confounds.

At last, we tested for an incentive effect on metacognitive sensitivity, a metric that measures the efficacy with which subjects discriminate between correct and incorrect answers using their confidence ratings (see Methods for details on its' computation). Replicating earlier findings[42], we found that incentive condition did not have a significant effect on metacognitive sensitivity ($F(2,62) = 0.25$, $p = 0.783$. Loss: $5.5973 ± 1.2106$, neutral: $4.8572 ± 1.0515$, gain: $5.2797 ± 0.8692$).

**fMRI results.** Having established the presence of a robust confidence-incentive interaction at the behavioral level, we next turned to the analysis of the functional neuroimaging data. Critically, our task allowed us to temporally distinguish the moment of stimulus presentation and choice—where the decision value and an implicit estimation of (un)certainty are expected to build up—from the incentive presentation and confidence rating moment—where the explicit, metacognitive confidence signal is expected to interact with the incentive (Fig. 2a, b).

*BOLD signal in the VMPFC correlates significantly with early certainty and incentives but weakly with confidence.* Our original

hypothesis proposes that incentives bias confidence because those two variables are both correlated to activity in the same brain area —presumably the VMPFC[22,23]. To test this hypothesis, we built a first fMRI GLM (GLM1) which modeled (1) early certainty during stimulus and choice, and (2) both incentives and confidence ratings during incentive/rating (Fig. 2c). Early certainty was defined and computed as the precursor of confidence (i.e., an incentive bias-free signal of confidence), that builds up before the commitment to a choice (see Methods for details). During choice, early certainty positively correlated with activation in the VMPFC and the posterior cingulate cortex (PCC) (Fig. 3a). This replicates several studies that have reported an early and automatic (i.e., without explicit instructions) encoding of confidence in the VMPFC[23,25,45]. Negative correlations of early certainty were observed in a widespread network including the bilateral dorsolateral prefrontal cortex (DLPFC) and rostro-lateral prefrontal cortex (RLPFC), bilateral anterior insula, right putamen, right inferior frontal gyrus, supplementary motor area, mid- and ACC, and bilateral inferior parietal lobe. This large network has already been implicated in uncertainty and metacognition[21].

During the incentive/rating moment, we found positive correlations between incentive value and activity in the VMPFC, extending to clusters in the dorsomedial prefrontal cortex (DMPFC) (Fig. 3b).

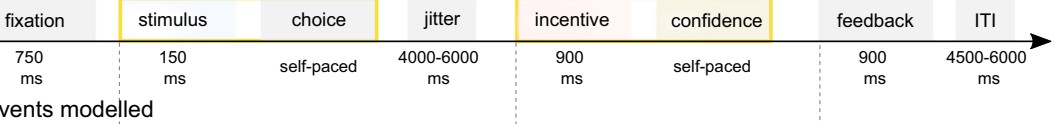

**a)** Events of interest

| fixation | stimulus | choice | jitter | incentive | confidence | feedback | ITI |

| 750 ms | 150 ms | self-paced | 4000-6000 ms | 900 ms | self-paced | 900 ms | 4500-6000 ms |

**b)** Events modelled

**c)** GLMs parametric regressors specification

| | | | |
|---|---|---|---|
| **GLM1:** | * early certainty (Z-scored) <br> *left/right choice (-1/1) | * incentive (-1/0/+1) <br> * confidence (Z-scored) | * accuracy (0/+1) |
| **GLM2a:** | * early certainty (Z-scored) <br> *left/right choice (-1/1) | * confidence (Z-scored) | * accuracy (0/+1) |
| **GLM2b:** | * early certainty (Z-scored) <br> *left/right choice (-1/1) | * incentive (-1/0/+1) <br> * early certainty (Z-scored) | * accuracy (0/+1) |
| **GLM3:** | * early certainty (Z-scored) <br> *left/right choice (-1/1) | * expected value (Z-scored) | * accuracy (0/+1) |
| **GLM4:** | * early certainty (Z-scored) <br> *left/right choice (-1/1) | * incentive (-1/0/+1) | * accuracy (0/+1) |

**Fig. 2 Overview of general linear models for fMRI analyses. a–b** Events of interest. The timeline depicts the succession of events within a trial. **a** Yellow boxes highlight the two events/timing of interest (stimulus/choice and incentive/confidence), that are modeled as stick function for the functional magnetic resonance imaging (fMRI) analysis. We also modeled the feedback event as a stick function. **c** General linear models (GLMs) parametric regressors specification. The graph displays the different combinations of parametric modulators of each event of interest for all GLMs used to analyze the fMRI data.

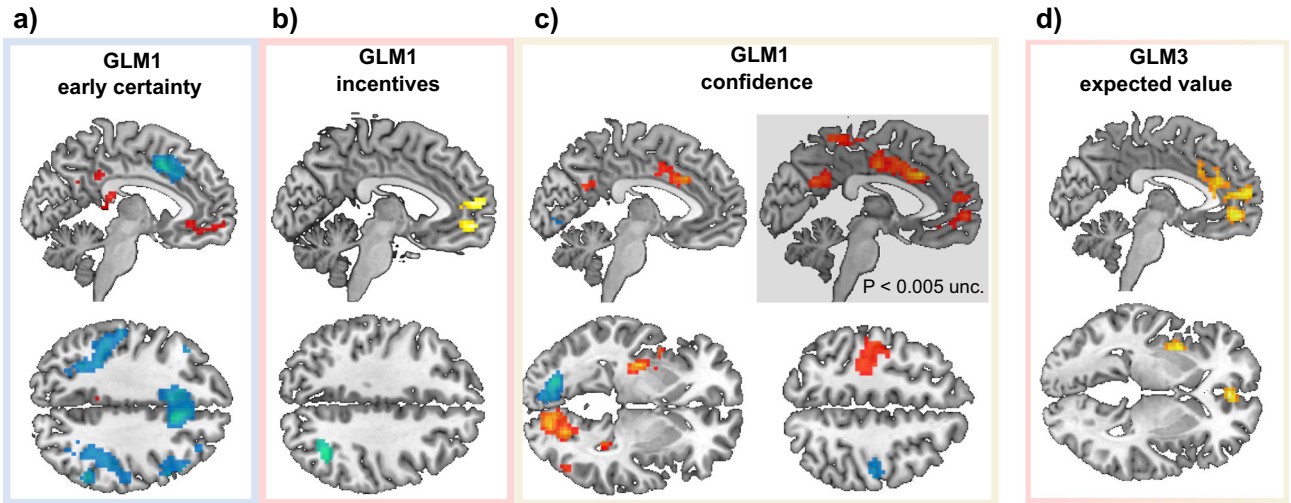

**a)** GLM1 early certainty **b)** GLM1 incentives **c)** GLM1 confidence P < 0.005 unc. **d)** GLM3 expected value

**Fig. 3 Whole-Brain fMRI Results. a–c** Whole-brain statistical blood-oxygen-level-dependent (BOLD) activity correlating with general linear model 1 (GLM1) "early certainty" (**a**), incentives (**b**), and confidence (**c**). **d** Whole-brain statistical maps of BOLD activity correlating with GLM3 "expected value". $N = 30$. Unless otherwise specified, all displayed clusters survived $p < 0.05$ family-wise error (FWE) cluster correction. Voxel-wise cluster-defining threshold was set at $p < 0.001$, uncorrected. Red/yellow clusters: positive activations. Blue clusters: negative activations. For whole-brain activation tables see Supplementary Data 1.

This is in line with our hypothesis and with a large body of neuro-economics literature[16]. A small cluster was detected in the occipital lobe, which negatively correlated with incentives.

Finally, regarding subjective confidence, we found significant positive effects in a large, lateralized visuo-motor network including the left primary motor cortex, left putamen, and left para-hippocampal gyrus, as well as the right cerebellum and right visual cortex (Fig. 3c). All those activations were mirrored in the negative correlation with confidence (although with lower and sometimes subthreshold significance), suggesting these brain

regions are part of the visuo-motor network that processes the movement of the cursor on the rating scale (remember that movements of the cursor were operationalized with the left (respective right) index finger to move the cursor toward the left (respective right).

Outside those visuo-motor areas, activity in a large cluster in the dorsal anterior cingulate cortex (dACC) and the mid-cingulate cortex (MCC) was found to positively correlate with confidence. Interestingly, an adjacent region of the dACC negatively correlated with early certainty in the choice period (Fig. 3a).

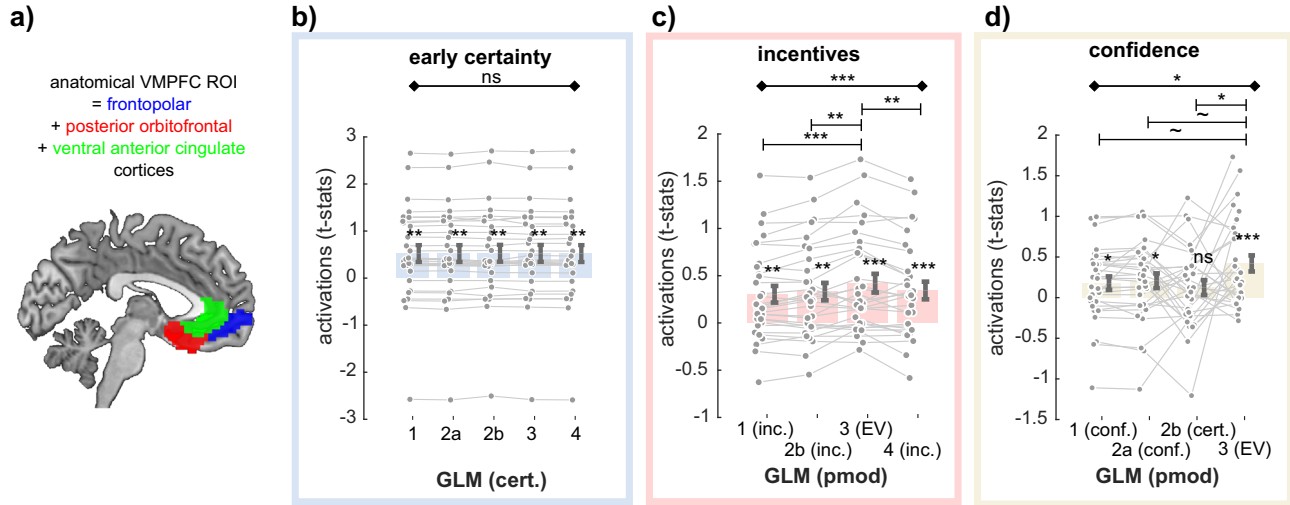

**Fig. 4 Activation in ventromedial prefrontal cortex across models. a** Anatomical ventromedial prefrontal cortex (VMPFC) region of interest (ROI). **b–d** Comparison of VMPFC activations to different specifications of early certainty during the choice moment (**b**), incentives during incentive/rating moment (**c**), and confidence during incentive/rating moment (**d**), as implemented in the different GLMs. Dots represent individual activations; bar and error bars indicate sample mean ± standard error of the mean. Gray lines highlight within-subject variation across the different specifications. N = 30. *Cert* early certainty; *Inc.* incentives; *conf.* confidence; *EV* expected value. Diamond-ended horizontal bars indicate the results of repeated-measure ANOVAs. Dash-ended horizontal bars indicate the result of post hoc paired t tests. ~p < 0.10; *p < 0.05; **p < 0.01; ***p < 0.001.

To our surprise, and in contradiction with our hypothesis, no whole-brain significant cluster was found in the VMPFC at our a priori defined statistical threshold. There were, however signs of subthreshold activations (Fig. 3c).

As observed with confidence activations, motor-related activity can be an important confound. To ensure that our activity patterns of interest (i.e., early certainty, incentive, and confidence) were not related to motor processes, we replicated our analyses using an exclusive motor-related mask, generated from large-scale auto-mated meta-analyses (see Methods for more details). Importantly, those control analyses revealed that most activations—with the exception of the visuo-motor activations identified in the confidence activation maps—remain significantly associated with our variables of interest (for whole-brain activation tables when using this exclusive mask, see Supplementary Data 2).

*Accounting for incentive bias in confidence does not restore VMPFC confidence activations.* Next, we attempted to understand the absence of strong correlations with confidence in the VMPFC, despite the same region robustly encoding early certainty and incentives (i.e., precursors of confidence). We reasoned that because confidence is biased by incentive, the shared variance between those two variables could have decreased our chances to reveal clear confidence signals during confidence ratings. We, therefore, built two control GLMs, which differed in how the incentive/rating period was modeled (Fig. 2c): GLM2a only included confidence as a parametric modulator, while GLM2b included incentive and early certainty (i.e., the precursor of confidence devoid of incentive shared variance). We defined an anatomical VMPFC ROI (see Methods and Fig. 4a), and extracted individual standardized regression coefficients (t values) corresponding to the confidence variable in those three GLMs (GLM1, GLM2a, GLM2b) (see Methods). We then tested whether the difference in the GLM specifications had an impact on these activations at the rating period (GLM1 and 2a: confidence; GLM2b: certainty) using repeated-measure analysis of variances (ANOVAs). Results showed that activations for GLM2a-confidence and GLM2b-early certainty during incentive/rating period were indistinguishable from GLM1-confidence (ANOVA, the main effect of GLMs: $F(2,29) = 0.68$; $p = 0.509$), falsifying the hypothesis that the weak confidence

activations in VMPFC observed with GLM1 were due to an ill-specified GLM.

*BOLD signal in the VMPFC strongly correlates with the EV.* Having established that BOLD activity in the VMPFC only weakly correlates with confidence after the incentive display, we proposed an alternative hypothesis—namely that the VMPFC encodes a signal commensurate to an EV. The rationale of this hypothesis is twofold. First, because confidence represents a subjective probability of being correct, it may be combined with information about the prospective monetary bonus to generate a representation of EV, once this reward information is revealed. Second, activity in the VMPFC has been repeatedly shown to correlate with EV in different contexts (lotteries, et cetera)[12–15]. To test this hypothesis, we built another fMRI GLM similar to the previous ones, but that instead modeled EV at the time of incentive/rating (GLM3; see Fig. 2c).

Whole-brain results showed massive positive correlations between EV and signal in the VMPFC stretching into the anterior medial prefrontal cortex, as well as the ventral and dorsal part of the ACC and the mid-cingulate cortex (Fig. 3d, Supplementary Data 1). There were no activation clusters negatively related to EV.

*BOLD signal in the VMPFC correlates better with EV than with other variables.* Although these results seem to validate our second hypothesis, our observation of more activations (wider cluster, lower p values) at the whole-brain level for EV than for confidence does not constitute a formal statistical test that VMPFC signals might rather correlate with EV than with confidence. These results may be owing to incentives and EV being highly correlated—in other words—, VMPFC activations to EV could simply be a result of VMPFC activations to incentives. To rule out these hypotheses, we built an additional GLM (GLM4), which only included incentive at the incentive/rating period (Fig. 2c). Again, we extracted VMPFC individual standardized regression coefficients (t values) corresponding to the early cer-tainty, incentive, and confidence-related activations in all avail-able GLMs. We tested whether the different specifications had an impact on those activations using repeated-measure ANOVAs,

**Table 1 Comparison of ventromedial prefrontal cortex (VMPFC) parametric activity (*t* values) as a function of model specification (GLMs).**

| Early certainty | GLM1 | GLM2a | GLM2b | GLM3 | GLM4 |
|---|---|---|---|---|---|
| | $0.52 \pm 0.18$ | $0.52 \pm 0.18$ | $0.53 \pm 0.18$ | $0.53 \pm 0.18$ | $0.52 \pm 0.18$ |
| | $t_{29} = 2.92$ | $t_{29} = 2.91$ | $t_{29} = 2.93$ | $t_{29} = 2.93$ | $t_{29} = 2.90$ |
| | $p = 0.007$ | $p = 0.007$ | $p = 0.007$ | $p = 0.007$ | $p = 0.007$ |
| RM ANOVA | - | - | - | - | - |
| $F(4,29) = 0.24$ | - | - | - | - | - |
| $p = 0.916$ | - | - | - | - | - |
| Incentive | GLM1 | | GLM2b | GLM3 | GLM4 |
| | $0.30 \pm 0.09$ | | $0.33 \pm 0.09$ | $0.42 \pm 0.10$ | $0.34 \pm 0.09$ |
| | $t_{29} = 3.45$ | | $t_{29} = 3.60$ | $t_{29} = 4.26$ | $t_{29} = 3.68$ |
| | $p = 0.002$ | | $p = 0.001$ | $p = 1.981 \times 10^{-4}$ | $p = 9.433 \times 10^{-4}$ |
| RM ANOVA | *t* test [3 vs 1] | | *t* test [3 vs 2b] | - | *t* test [3 vs 4] |
| $F(3,29) = 10.67$ | $0.12 \pm 0.03$ | | $0.09 \pm 0.03$ | - | $0.08 \pm 0.03$ |
| $p = 4.837 \times 10^{-6}$ | $t_{29} = 3.90$ | | $t_{29} = 3.38$ | | $t_{29} = 2.97$ |
| | $p = 5.306 \times 10^{-4}$ | | $p = 0.002$ | | $p = 0.006$ |
| Confidence | GLM1 | GLM2a | GLM2b | GLM3 | |
| | $0.18 \pm 0.08$ | $0.21 \pm 0.09$ | $0.12 \pm 0.09$ | $0.42 \pm 0.10$ | |
| | $t_{29} = 2.14$ | $t_{29} = 2.30$ | $t_{29} = 1.35$ | $t_{29} = 4.26$ | |
| | $p = 0.041$ | $p = 0.028$ | $p = 0.187$ | $p = 1.981 \times 10^{-4}$ | |
| RM ANOVA | *t* test [3 vs 1] | *t* test [3 vs 2a] | *t* test [3 vs 2b] | - | |
| $F(3,29) = 3.22$ | $0.24 \pm 0.13$ | $0.21 \pm 0.12$ | $0.30 \pm 0.13$ | - | |
| $p = 0.027$ | $t_{29} = 1.92$ | $t_{29} = -1.72$ | $t_{29} = 2.36$ | - | |
| | $p = 0.064$ | $p = 0.096$ | $p = 0.025$ | - | |

The table reports descriptive and inferential statistics on VMPFC region of interest (ROI) parametric activations with three different variables of interest: early certainty effects at the choice moment, incentive effects at rating moment, and confidence effects at rating moment (see Fig. 4). Per effect of interest, results of one-sample *t* tests against zero, repeated-measure (RM) ANOVAs on the main effect of GLMS, and post hoc *t* test results are shown.

and post hoc *t* tests (Fig. 4, Table 1). Although activations for early certainty during choice moment were similar for all GLMs (ANOVA, main effect of GLM; $F(4,29) = 0.24$, $p = 0.916$; Fig. 4b), GLM specification had an impact on both the incentive activations (ANOVA, main effect of GLM; $F(3,29) = 10.67$, $p = 4.837 \times 10^{-6}$; Fig. 4c) and the confidence activations (ANOVA, main effect of GLM; $F(3,29) = 3.22$, $p = 0.027$; Fig. 4d) during incentive/rating moment. In both cases, post hoc *t* tests showed that *t* values extracted from the GLM3 that related to the EV regressor were significantly higher than from other GLMs with a different coding of incentives (GLM1 vs GLM3: $t_{29} = 3.90$, $p = 5.306 \times 10^{-4}$; GLM2b vs GLM3: $t_{29} = 3.38$, $p = 0.002$, GLM4 vs GLM3: $t_{29} = 2.97$, $p = 0.006$), and marginally higher from other GLMs with a different coding of confidence (GLM1 vs. GLM3: $t_{29} = 1.92$, $p = 0.064$; GLM2a vs. GLM3: $t_{29} = 1.72$, $p = 0.096$; GLM2b vs. GLM3: $t_{29} = 2.36$, $p = 0.025$). Overall these analyses suggest that the VMPFC combines incentive and confidence signals in the form of an EV signal.

*Qualitative falsification of the EV model of VMPFC activity.* At last, in order to confirm the conclusions drawn from our quantitative comparison of VMPFC activations, we ran a qualitative falsification exercise[46]. Leveraging the factorial design of our experiment, we could draw qualitative patterns of activations that would be expected under different hypotheses underlying VMPFC activation (Fig. 5a).

To this end, we designed a final GLM (GLM5) that divided the task into two timepoints (stimulus/choice and incentive/rating), and three incentive conditions, and that incorporated a baseline and a regression slope with confidence judgment for all these events. We then extracted the VMPFC activations for all these regressors using our ROI, and compared them with the theorized qualitative patterns we would expect if the VMPFC encoded one of these variables (Fig. 5b, c and Table 2, Table 3). As expected, at the moment of the stimulus/choice, there was no effect of incentive conditions on

VMPFC baseline activity, nor on its correlation with confidence —"slope" (ANOVA baseline: $F(2,29) = 0.36$, $p = 0.701$; ANOVA correlation with confidence: $F(2,29) = 0.56$, $p = 0.574$). Basically, the slopes were significantly positive in all three incentive conditions (Loss: $t_{29} = 2.10$, $p = 0.045$; Neutral: $t_{29} = 2.43$, $p = 0.021$; Gain: $t_{29} = 3.04$, $p = 0.005$), confirming that the VMPFC encodes an early certainty signal.

At rating moment, incentive conditions had an effect on both VMPFC baseline activity, and on the correlation of VMPFC activity with confidence (ANOVA baseline: $F(2,29) = 8.56$, $p = 5.543 \times 10^{-4}$; ANOVA correlation with confidence: $F(2,29) = 5.26$, $p = 0.008$). Post hoc testing revealed that VMPFC baseline activity was significantly larger in gain versus loss ($t_{29} = 3.47$, $p = 0.002$) and in gain versus neutral conditions ($t_{29} = 3.17$, $p = 0.004$), but not in neutral versus loss condition ($t_{29} = 0.43$, $p = 0.673$) (see Table 3). This constitutes a deviation from a standard linear model of incentives, and suggest that different regions might process incentives in gains and loss contexts[47].

Moreover, we found that the correlation of VMPFC activity with confidence is significantly positive in the gain condition only ($t_{29} = 3.29$, $p = 0.003$), and not in the loss ($t_{29} = -0.75$, $p = 0.457$) nor neutral ($t_{29} = 0.70$, $p = 0.491$) conditions. The correlation with confidence was therefore significantly higher in gain versus loss ($t_{29} = 3.13$, $p = 0.004$) and in gain versus neutral conditions ($t_{29} = 2.02$, $p = 0.053$), but not in neutral versus loss condition ($t_{29} = 1.03$, $p = 0.313$). Although the absence of correlation in the neutral condition would be expected if the VMPFC encodes EV, the lack of correlation in the loss condition was not predicted by any of our models (Fig. 5a). Because VMPFC confidence activations were robustly observed in the gain domain, as well as VMPFC early certainty activations in all three conditions, we suggest that the lack of VMPFC confidence activations in the neutral and loss conditions is a feature of the VMPFC signal, rather than a failure of our design to elicit those activations (e.g., due to limited statistical power or excessive statistical noise).

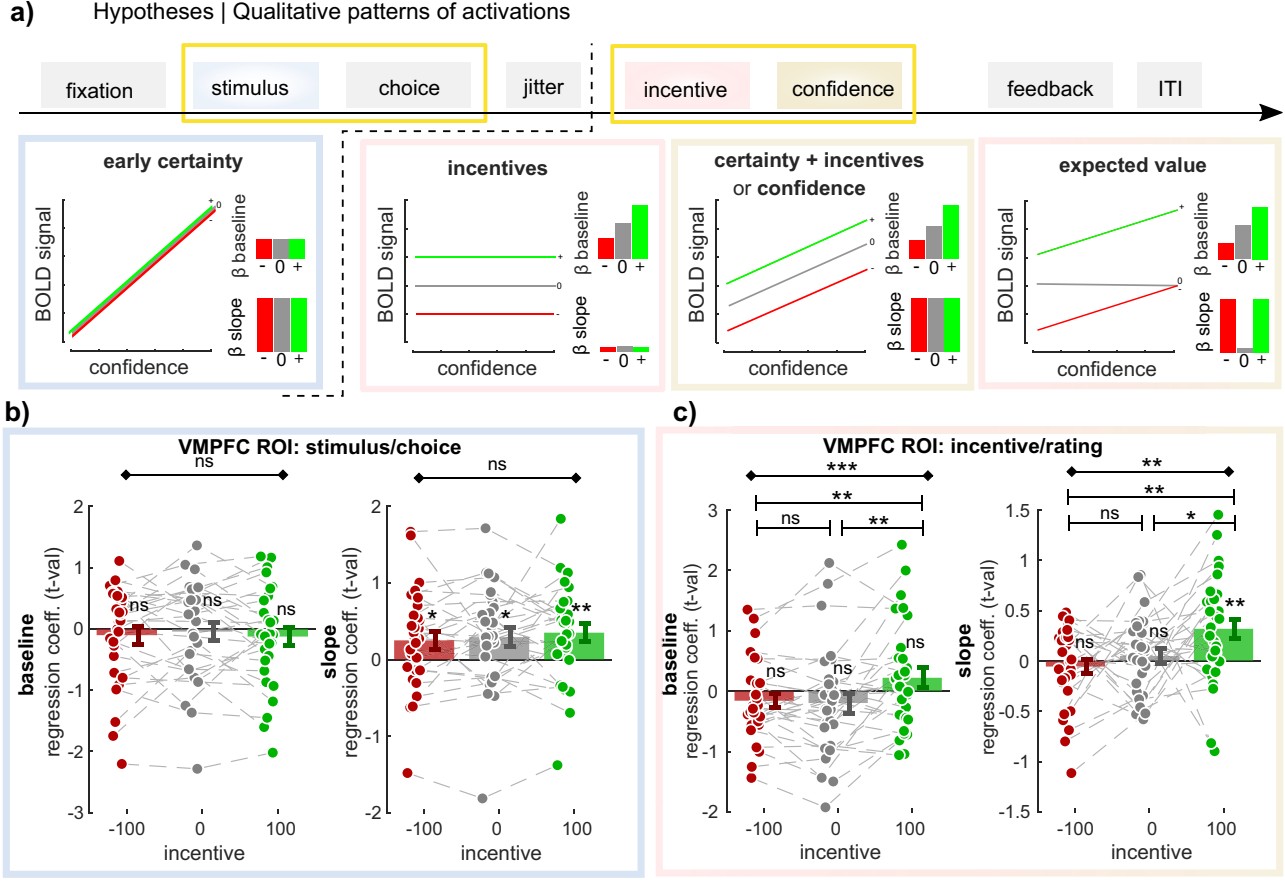

**Fig. 5 Activation in ventromedial prefrontal cortex across incentives and time points. a** Qualitative ventromedial prefrontal cortex (VMPFC) activation patterns predicted under different models. The different boxes present how blood-oxygen-level-dependent (BOLD) signal should vary with increasing confidence in the three incentive conditions (green: +100; gray: 0; red: −100), under different hypotheses (i.e., encoding different variables), at different time points. Bar graphs in insets summarize these relationships as expected intercepts (or baseline—top) and slope (bottom). **b**–**c** VMPFC region of interest (ROI) analysis (N = 30). T values corresponding to baseline and regression slope were extracted in the three incentive conditions, and at the two time points of interest (**b**: stimulus/choice; **c**: incentive/rating). Dots represent individual activations; bar and error bars indicate sample mean ± standard error of the mean. Gray lines highlight within-subject variation across the different incentive conditions. Diamond-ended horizontal bars indicate the results of repeated-measure ANOVAs. Dash-ended horizontal bars indicate the result of post hoc paired t tests. ns: P > 0.05; *P < 0.05; **P < 0.01; ***P < 0.001.

**Table 2 Comparison of ventromedial prefrontal cortex (VMPFC) activity at the choice moment (t values), as a function of incentive condition.**

| Choice/stim | Baseline | Inc. −100 | Inc. 0 | Inc. +100 | RM ANOVA |
|---|---|---|---|---|---|
| | | $-0.10 \pm 0.15$ | $-0.04 \pm 0.15$ | $-0.13 \pm 0.15$ | $F(2,29) = 0.36$ |
| | | $t_{29} = -0.70$ | $t_{29} = -0.30$ | $t_{29} = -0.85$ | $p = 0.701$ |
| | | $p = 0.490$ | $p = 0.770$ | $p = 0.400$ | |
| | Slope | Inc. −100 | Inc 0 | Inc. +100 | RM ANOVA |
| | | $0.250 \pm 0.12$ | $0.29 \pm 0.12$ | $0.35 \pm 0.12$ | $F(2,29) = 0.56$  $p = 0.576$ |
| | | $t_{29} = 2.10$ | $t_{29} = 2.43$ | $t_{29} = 3.04$ | |
| | | $p = 0.045$ | $p = 0.021$ | $p = 0.005$ | |

The table reports descriptive and inferential statistics on VMPFC region of interest (ROI) parametric activations in our three incentive conditions during the choice moment, for both baseline activity as well as the correlation with early certainty (i.e., slope) (see Fig. 5B). Results of repeated-measures (RM) ANOVAs and one-sample t tests against 0 are shown. Inc. = incentive.

To evaluate whether the lack of robust confidence activation in the neutral and loss condition could be caused by the rough averaging of the VMPFC signal over the anatomical ROI, we also performed a finer-grained analysis. We extracted confidence activations in the three conditions and two timepoints at the voxel-level in a large anatomical area covering most of the medial prefrontal cortex, averaged those activations over two dimensions (respectively X and Z, and X and Y), and assessed how activations unfold over the last dimension—respectively Y and Z (Fig. 6). This

last analysis confirmed three main facts: first, the early certainty activations are robustly observed in the same portion of the VMPFC, and—as expected—with similar effect sizes in the three conditions; second, the confidence activations in the gain condition are observed at similar levels as the early certainty activations, confirming that our experimental design elicits robust activations at the incentive/ confidence rating time-point; third, no confidence activations can be detected at this finer-grained level in the neutral or loss condition, in the VMPFC. If anything, it seems that the confidence activations in

**Table 3 Comparison of ventromedial prefrontal cortex (VMPFC) activity at rating moment (*t* values), as a function of incentive condition.**

| Incentive/ratinbg | Baseline | Inc. −100 | Inc. 0 | Inc. + 100 | RM ANOVA |
|---|---|---|---|---|---|
| | | −0.16 ± 0.12 | −0.20 ± 0.17 | 0.22 ± 0.16 | $F(2,29) = 8.56$ $p = 5.543 \times 10^{-4}$ |
| | | $t_{29} = -1.31$ | $t_{29} = -1.19$ | $t_{29} = 1.37$ | |
| | | $p = 0.20$ | $p = 0.25$ | $p = 0.18$ | |
| | | $t$ test [−100 vs 0] | $t$ test [0 vs 100] | $t$ test [−100 vs 100] | |
| | | 0.04 ± 0.09 | −0.42 ± 0.13 | −0.38 ± 0.11 | |
| | | $t_{29} = 0.43$ | $t_{29} = 3.17$ | $t_{29} = 3.47$ | |
| | | $p = 0.673$ | $p = 0.004$ | $p = 0.002$ | |
| | Slope | Inc. −100 | Inc. 0 | Inc. + 100 | RM ANOVA |
| | | −0.06 ± 0.07 | 0.05 ± 0.07 | 0.32 ± 0.10 | $I(2,29) = 5.26$ $p = 0.008$ |
| | | $t_{29} = -0.75$ | $t_{29} = 0.70$ | $t_{29} = 3.29$ | |
| | | $p = 0.457$ | $p = 0.491$ | $p = 0.003$ | |
| | | $t$ test [−100 vs 0] | $t$ test [0 vs 100] | $t$ test [−100 vs 100] | |
| | | −0.11 ± 0.10 | −0.27 ± 0.13 | −0.38 ± 0.12 | |
| | | $t_{29} = 1.03$ | $t_{29} = 2.02$ | $t_{29} = 3.13$ | |
| | | $p = 0.313$ | $p = 0.053$ | $p = 0.004$ | |

The table reports descriptive and inferential statistics on VMPFC region of interest (ROI) parametric activations in our three incentive conditions during rating moment, for both baseline activity as well as the correlation with confidence (i.e., slope) (see Fig. 5C). Results of one-sample *t* tests against 0, repeated-measures (RM) ANOVAs, and post hoc *t* tests are shown. Inc. = incentive.

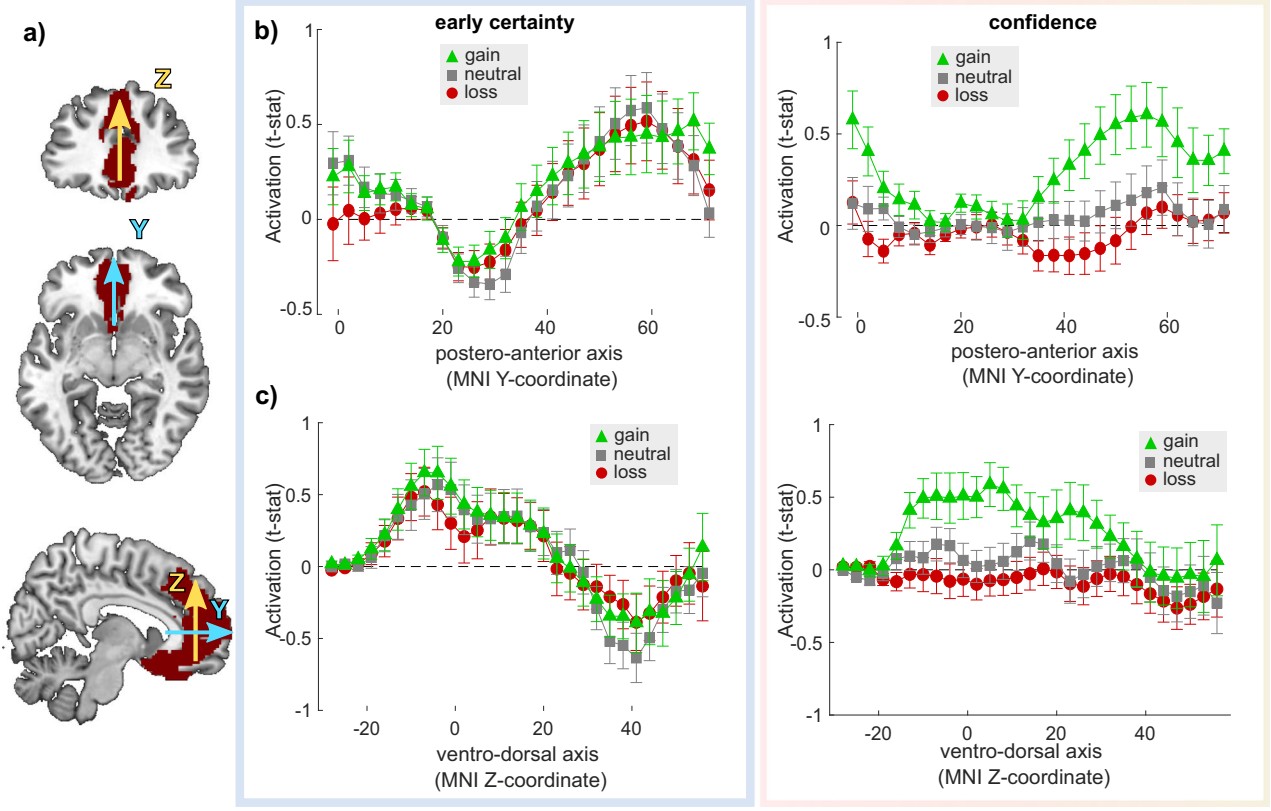

**Fig. 6 Activation in ventromedial prefrontal cortex across Y and Z dimensions. a** Large anatomical medial prefrontal cortex region of interest (ROI). The Y (blue) and Z (yellow) arrows indicate the dimensions over which the signal is extracted and marginalized—respectively, corresponding to the postero-anterior axis and ventro-dorsal axis. **b–c** MPFC region of interest (ROI) analysis of confidence activations, at the voxel-level, marginalized over the Y (**b**) and Z (**c**) dimensions. Voxel-wise *T* values corresponding to regression slope were extracted in the three incentive conditions (green: +100; gray: 0; red: −100), and at the two time points of interest (left: stimulus/choice; right: incentive/rating), averaged over two dimensions and plotted as a function of the third dimension. Dots and error bars indicate sample mean ± standard error of the mean (*N* = 30).

the loss condition trend toward a negative correlation between VMPFC BOLD signal and confidence.

Overall, these results initially explain why EV appears a better model of VMPFC activation than confidence and/or incentive (correct pattern in gains and neutral conditions), but ultimately falsify this account by demonstrating the absence of positive correlation between VMPFC activation and confidence in the loss condition.

## Discussion

In this study, we set out to investigate the neural signature of incentive bias on confidence estimations, using an fMRI-optimized

version of an incentivized perceptual decision-making task[42]. First, at the behavioral level, we replicated the biasing effect of incentives on confidence estimation, in the form of higher confidence in gain contexts and lower confidence in loss context, despite equal difficulty and performance. This result is the fourth independent replication of this bias, initially revealed in perceptual decision making and later generalized in a reinforcement-learning task[43,44]. Note, however, that the bias' effect size remains small—a few average confidence percentage points at the population level—which a priori limits our ability to dissect its precise neurophysiological basis with current (correlational) functional neuroimaging techniques.

Our initial goal and hypothesis were therefore quite simple and modest. In the literature, it is now well established that the BOLD signal in the VMPFC correlates with confidence and/or values in a variety of tasks[22–25,29,45]. We reasoned that if we could provide evidence for the presence of both incentive and confidence signals in the VMPFC during our task, this would reinforce the intuition that the VMPFC has a role in the observed behavioral phenomenon, i.e., the incentive bias on confidence. Our neuroimaging predictions were that (1) the VMPFC should correlate with early certainty before and during choice, regardless of the context, and (2) the VMPFC should integrate confidence and incentive after the choice and the revealing of the incentive condition. Our broader, speculative neural hypothesis was that during this last confidence judgment step, a third-party metacognitive region or network would sample signal in the VMPFC[48,49], and incidentally end up with a biased confidence estimate incorporating incentive signal. Our limited sample size combined with some known limits of brain-behavior analyses[50] restricted a priori any ambition to validate a neurobiological model of the observed confidence bias by running inter-individual correlations between VMPFC activations and the confidence bias estimated at the behavioral level.

Our fMRI investigation of the neural correlates of early certainty confirms our first prediction: BOLD activity in the VMPFC positively correlates with early certainty in all conditions. This result replicates and extends previous studies demonstrating this area to be associated to the initial and automatic processing of confidence during choice[22,23,25]. In parallel with this positive correlation in the VMPFC, we also observed widespread negative correlations in the DLPFC, DMPFC, and insula, a network robustly associated with both metacognition and uncertainty[21,29,51]. Contrary to our second prediction, we only found weak evidence (i.e., at a lower statistical threshold than the one we defined a priori) for confidence encoding in the VMPFC. Robust activations were nonetheless observed in the dACC, a region known to be recruited in metacognitive judgments[20,52].

Given that the lack of robust confidence signal in the VMPFC is somewhat in contradiction with what we expected from our previous work, as well as numerous other reports in the literature[22–25,29,45], we formulated an alternative hypothesis: we proposed that VMPFC could encode a signal commensurate to an expected reward (or EV), i.e., incorporating the subjective probability of being correct with the potential incentive bonus when revealed. Whole-brain activations and ROI quantitative analyses clearly showed that this second hypothesis seems to give a better account of VMPFC BOLD activations. EV signals are frequently reported in the VMPFC, but mostly in reinforcement-learning contexts, where they are critical to both choices between available options and learning—i.e., value updating, through the computation of prediction errors[53]. In the present perceptual task, there is no learning, therefore no explicit need to encode EV.

Because quantitative comparisons of hypotheses are notoriously hard to interpret, we decided to leverage the factorial aspect of our design to proceed to a qualitative hypothesis falsification, to validate—or falsify—the EV account of VMPFC activity[46]. In short, different hypotheses about what should be contained in VMPFC signal (EV, confidence, and/or incentives) predict different patterns of activations (baseline and correlation with confidence) in our different incentive conditions. From activity extracted from an anatomical VMPFC ROI, it is clear that VMPFC activity correlates with confidence only in the gain context, once the incentive has been revealed. This finding explains why the EV hypothesis obtained stronger quantitative support than the confidence and/or incentives hypotheses (as the VMPFC activity pattern is similar to the EV predictions in the gain and neutral context). However, it also ultimately falsifies this EV hypothesis as well, as VMPFC activity does not seem to correlate with confidence in the loss context. Interestingly, VMPFC does correlate with early certainty—a precursor of confidence—in all conditions before the incentives are revealed. Therefore, it does not seem that the VMPC fails to activate in the neutral and loss conditions, but rather that the signal is actively suppressed once those contexts are explicit. Moreover, the fact that we do not observe confidence activations in neutral or loss conditions is also not due to the fact that participants are less focused on evaluating confidence in those conditions compared to the gain condition, as we showed that the confidence sensitivity is identical in all incentive conditions. In summary, we believe that our results show a complex picture of disruptions of confidence signals within the VMPFC in response to motivational signals.

The absence of VMPFC confidence signal in the neutral condition might seem at odds with other studies that report such signal in non-incentivized tasks such as pleasantness or desirability ratings[23]. One possible explanation is that VMPFC confidence signals, like attentional modulation of evidence integration[54], are primarily observed for behavior or conditions that are relevant to participants' goals: in non-incentivized tasks such as pleasantness or desirability ratings, participants still have a goal, which is to provide ratings that are as accurate as possible. In our task, if the goal of participants is to maximize their score, the neutral condition might not be goal-relevant, which could result in a disrupted VMPFC confidence signal. Note that because our design features interleaved (rather than blocked) conditions, the valence manipulation is somewhat exacerbated, as the succession of the different conditions limit the contextualization of outcomes (whereby the absence of loss could be reframed as a relative gain in a loss-block). Also, because trials featuring gains, losses, and neutral incentives follow each-others in a pseudorandomized order, the interleaved design also prevent any systematic bias or confound for the valence effects (at the behavioral or neurobiological levels) that could be due to the processing of the feedbacks (gains, losses, or nothing).

The notion that there are different brain networks that execute symmetric computations in gains versus loss contexts is increasingly popular[47,55]. Because the positive, gain context network also typically includes the VS (see e.g.,[12,16] we replicated all analyses using an anatomical VS ROI (see Supplementary Note 2). These analyses qualitatively rendered very similar results to what we observed in the VMPFC. In the present data set though, we did not find any region correlating either positively or negatively with confidence in the loss context, even when exploring the whole-brain level with very lenient statistical thresholds. The dACC is a promising area, since it has repeatedly been associated with loss anticipation and correlated positively with subjective confidence in our data. However, when we performed a similar falsification exercise within the dACC as we used within the VMPFC (see Supplementary Note 3), the results were similar to the VMPFC activation patterns: dACC activity only correlated with confidence within the gain contexts. In summary, it remains an open question what the neurobiological correlates of confidence judgments in loss contexts are.

Our results constitute a stepping stone and have important implications for studying clinical populations where these (meta) cognitive processes go awry. It shows that motivational processes can influence confidence, and when there are discrepancies between one's behavior and confidence in that behavior, this could give rise to pathological decision making. Indeed, several psychiatric disorders such as addiction, obsessive-compulsive disorder, and schizophrenia have been associated with disrupted incentive processing[56–60] and studies have additionally demonstrated distorted confidence estimations in these groups[61]. Our study indicates that the VMPFC is a key region involved in the interaction between motivation and metacognition, and VMPFC function is also often affected in many psychiatric disorders[62]. The current study provides a means of studying neurobiological explanations for confidence abnormalities and their interaction with incentive motivation in the clinical population which can potentially impact clinical practice, as it could help treat psychopathology[62]. Therefore, the relationship between motivational processes and confidence estimation and their role in psychopathology warrants future investigation.

In conclusion, we show that although the VMPFC seems to encode both value and metacognitive signals, these metacognitive signals are only present during the prospect of gain and are disrupted in a context with loss or no monetary prospects. Studies targeting this problem within a finer spatial[24,63,64] and/or temporal scale[65] could help with resolving and better comprehending biased confidence judgments and metacognition overall.

## Methods

**Participants**. We included 33 right-handed healthy participants with normal or corrected to normal vision. Exclusion criteria were an IQ below 80, insufficient command of the Dutch language, or MRI contraindications. All experimental procedures were approved by the Medical Ethics Committee of the Academic Medical Center, University of Amsterdam (METC 2015_319), and participants gave written informed consent. Participants were compensated with a base amount of €40 and additional gains based on task performance. Session-level behavioral and fMRI data were excluded when task accuracy was below 60% or when subjects did not show sufficient variation in their confidence reports (standard deviation of confidence judgments < 5 confidence points), and session-level fMRI data when participants showed head movements > 3.5 mm. This led to the inclusion of 32 participants (18/14 females/males, 18–58 years old (sd: 9.76)) for the behavioral analyses and 30 for the fMRI analyses, of which four participants contributed only one of two task sessions.

**Decision-making and confidence judgment task**. We adapted the task from Lebreton et al.[42] for use in an fMRI environment with fMRI suitable timing intervals. For an overview and details, see Fig. 1a. All tasks used in this study were implemented using MATLAB® (MathWorks Inc., Sherborn, MA, USA) and the COGENT toolbox (www.vislab.ucl.ac.uk/cogent.php).

**Study procedure**. On the day of testing, subjects were first assessed for clinical and demographic data, after which they performed one practice session (10 trials) outside of the scanner and another one inside the scanner to become acquainted with the task. Subjects were instructed that they would only be rewarded based on their performance (i.e., they should be as accurate as possible to maximize their earnings), and that it was important to give accurate confidence judgments. They were notified that 50% confidence would signal that they made a guess, whereas 100% confidence would signal that they were absolutely certain that they made the correct choice. Thus, performance but not confidence was incentivized. According to our previous findings[42], this design elicits incentive bias on confidence while keeping confidence sensitivity identical across conditions—an important consideration when interpreting differences in confidence activations between those conditions. All subjects initially performed a 144-trial calibration session inside the scanner to tailor the difficulty levels of the task to each individual and to keep performance constant across subjects. This was done using a staircase procedure, in which data were used to estimate a full psychometric function, whose parameters were used to generate stimuli for the main task, spanning three difficulty levels (i.e., 65%, 75%, and 85% accuracy, on average) (for details, see ref.[42]).

Two sessions of the main task were performed in the fMRI scanner, each consisting of 72 trials with 24 trials per incentive condition, presented in random order. The practice task, calibration, and main sessions were projected onto an Iiyama monitor in the fMRI environment, which subjects could see through a 45-degree angle mirror fixed to the head coil. After completing the fMRI task, six

random trials were drawn (i.e., two of each incentive condition) on which the payment was based. If subjects made an accurate choice, they would either gain or avoid losing points, whereas they would miss out on gaining or losing points when making an error. In the neutral trials, nothing was at stake. Finally, the total amount of points were converted to money.

**Behavioral measures**. We extracted various trial-by-trial experimental factors (evidence, incentive, and difficulty level) and behavioral measures (accuracy, subjective confidence ratings, RTs). Control analyses were performed to confirm the properties of confidence ratings (Supplementary Note 4). Three additional variables were computed as combinations of those experimental factors and behavioral measures: early certainty, EV, and metacognitive sensitivity.

*Early certainty*. We built an "early certainty" variable that represents a confidence signal prior to the biasing effects of incentives. We assume that such an early certainty signal should be encoded automatically at the moment of choice, in turn allowing us to investigate confidence signals with and without incentive bias[23]. Importantly, such a signal should be highly correlated with the later, biased confidence judgment obtained from the subjects, while exhibiting no statistically significant relationship with incentives. Therefore, we used a leave-one-trial-out approach to obtain trial-by-trial estimations of early certainty[52]. We fitted a generalized linear regression model to each subject's subjective confidence ratings using choice and stimulus features as predictors (i.e., log-transformed RTs, evidence, accuracy, and the interaction between accuracy, and evidence), using the whole individual dataset but trial X. We then applied this model's estimates to generate predictions about the early certainty in trial X, using the choice and stimulus features of trial X. This process was repeated for every trial, resulting in a trial-by-trial prediction of early certainty based on stimulus features at choice moment. The resulting early certainty signal featured high correlation with confidence, and no statistical relationship with incentives (see Supplementary Note 5 for more details). Importantly, since the early certainty signal follows the main properties of confidence judgments (Supplementary Fig. 6), but does not show any incentive bias, this critically enables us to differentiate between non-biased confidence signals during decision-making and biased confidence signals after incentivization.

*EV*. We computed a value-based measure of EV. In our task paradigm, EV was computed as an integrative signal of early certainty (i.e., the non-biased probability of being correct) and the incentive value (i.e., the value-context of the current trial). Early certainty ratings represent the subjects' probability of being correct, and thus the probability of gaining (or avoid losing) the incentive at stake. Thus, EV corresponds to 0 in the neutral condition (no value is expected to be gained or lost), is equal to early certainty in the gain condition (e.g., being 100% certain results in a maximal EV in a positive incentive environment), and is equal to early certainty—100 (e.g., being 100% certain in a loss trial results in an EV of 0, as you avoid losing).

*Metacognitive sensitivity*. Metacognitive sensitivity is a metric that indicates how well an observer's confidence judgments discriminate between their correct and incorrect answers and can be represented using several indexes. For example, discrimination is a metric calculated as the difference between the average confidence for correct answers and the average confidence for incorrect answers, whereas meta-d' is a metric based on the Signal Detection Theory framework[66]. Notably, meta-d' computations are known to be imprecise in designs with a low number of trials per condition[67]. This, together with results from our earlier work[42] showing high correlations between discrimination and meta-d', as well as identical conclusions with respect to the effects of incentives on these measures, lead to us using the discrimination metric as our measure of metacognitive sensitivity.

**fMRI acquisition and preprocessing**. fMRI data were acquired by using a 3.0 Tesla Intera MRI scanner (Philips Medical Systems, Best, The Netherlands). Following the acquisition of a T1-weighted structural anatomical image, 37 axial T2*-weighted EPI functional slices sensitive to BOLD contrast were acquired. A multi-echo (three echoes) combine interleaved scan sequence was applied, designed to optimize functional sensitivity in all parts of the brain[68]. The following imaging parameters were used: repetition time (TR), 2.375 seconds; echo times (TEs), 9.0 ms, 24.0 ms, and 43.8 ms, (total echo train length: 75 ms); 3 mm (isometric) voxel size; 37 transverse slices; 3 mm slice thickness; 0.3 mm slice-gap. Two experimental sessions were carried out, each consisting of 570 volumes. All further analyses were performed using MATLAB® with SPM12 software (Wellcome Department of Cognitive Neurology, London, UK).

Raw multi-echo functional scans were weighed and combined into 570 volumes per scan session. During the combining process, realignment was performed on the functional data by using linear interpolation to the first volume. The first 30 dummy scans were discarded. The remaining functional images were co-registered with the T1-weighted structural image, segmented for normalization to Montreal Neurological Institute (MNI) space, and smoothed using a Gaussian kernel of 6 mm at full-width at half-maximum.

Owing to sudden motion, in combination with the interleaved scanning method, a number of subjects showed artifacts in some functional volumes. In order to reduce those artifacts, the Art-Repair toolbox[69] was used to detect large volume-to-volume movement and repair outlier volumes. The toolbox identifies outliers by using a threshold for the variation of the mean intensity of the BOLD signal and a volume-to-volume motion threshold. A threshold of 1.5% variation from the mean intensity was used to detect and repair volume outliers by interpolating from the adjacent volumes ($n = 12$).

**Statistics and reproducibility: behavioral analyses**. All behavioral analyses were performed using MATLAB® and the R environment (RStudio Team (2015). RStudio: Integrated Development for R. RStudio, Inc., Boston, MA). For the statistical analyses reported in the main text, we used linear mixed-effects models (estimated with the fitglme function in MATLAB®) to model accuracy, RTs, and confidence. In order to analyze the effect of the incentive condition (i.e., of our experimental manipulation of incentives), for all three trial-by-trial dependent variables we used the absolute incentive value (i.e., the absolute value of the monetary incentive, $|V|$, coded as 0 and $+1$) and the net incentive value (i.e., the linear value of the monetary incentive, V, coded as $-1$, 0, and $+1$) as predictor variables. All mixed models included random intercepts and random slopes ($N = 32$). Additional control analyses are reported in Supplementary Note 4. For the analysis of metacognitive sensitivity, we performed a repeated-measures ANOVA, with net incentive value as within-subject factor.

**Statistics and reproducibility: fMRI analyses**. All fMRI analyses were conducted using SPM12. All general linear models (GLMs) were estimated on subject-level ($N = 30$) with two moments of interest: the moment of choice (i.e., presentation of the Gabor patches) and the moment of incentive presentation/confidence rating (Fig. 2). The rating moment follows the presentation of the incentive after 900 ms, hence the decision to analyze them as a single moment of interest. Moreover, the GLMs also included a regressor for the feedback moment, which was not of interest for analysis, but was intended to explain variance in neural responses related to value and accuracy feedback, but unrelated to the decision-making process.

When using parametric modulators in our GLMs, those were not orthogonalized and competed to explain variance. Nuisance regressors consisting of six motion parameters were included in all GLMs. Regressors were modeled separately for each scan session and constants were included to account for between-session differences in mean activation. All events were modeled by convolving a series of delta functions with the canonical hemodynamic response function at the onset of each event and were linearly regressed onto the functional BOLD-response signal. Low-frequency noise was filtered with a high pass filter with a cutoff of 128 seconds. All contrasts were computed at subject-level and taken to a group-level mixed-effect analysis using one-sample $t$ tests.

We controlled for the number of sessions while making the first-level contrasts. We assessed group-level main effects by applying one-sample $t$ tests against 0 to these contrast images. All whole-brain activation maps were thresholded using FWE for multiple corrections at cluster level ($p_{FWE\_clu} < 0.05$), with a voxel cluster-defining threshold of $p < 0.001$ uncorrected.

*GLM1: neural signatures of certainty, incentive, and confidence*. GLM1 consisted of three regressors for the three moments of interest: "choice", "incentive/rating", and "feedback", to which one or more parametric modulators (pmod) were added (Fig. 2). The regressors were specified as stick function time-locked to the onset of the events. The choice regressor was modulated by two pmods: early certainty ($z$ scored before entering the GLM) and button press (left/right choice) in order to control for activity related to motor preparation. The incentive/rating regressor was modulated by two pmods: incentive value and subjective confidence level ($z$ scored). At last, the feedback regressor was modulated by a pmod of accuracy.

Importantly, to ensure that our brain activations of interest (i.e., related to early certainty, incentive, and confidence) were not confounded by motor-related activations, we performed control analyses that implemented exclusive masking for motor activations. To do so, we generated the exclusive mask from "Neurosynth" (a platform for large-scale, automated synthesis of fMRI data[70]), using the term 'motor' (https://neurosynth.org/analyses/terms/motor/). This mask represents key regions related to motor processes as identified by an automated meta-analysis of 2565 studies.

*GLM2a: control for incentive bias 1*. GLM2a consisted of the same regressors as GLM1, except that the rating moment was only modulated by confidence judgments (i.e., we deleted the incentive modulator).

*GLM2b: control for incentive bias 2*. GLM2b consisted of the same regressors as GLM1, except that the pmod of confidence judgments at the rating moment was replaced by a pmod for early certainty.

*GLM3: neural signatures of EV*. GLM3 consisted of the same regressors as GLM1, except that rating moment was modulated by a single pmod of EV.

*GLM4: control for incentive*. GLM4 consisted of the same regressors as GLM1, except that the rating moment was only modulated by incentives (i.e., we deleted the confidence judgment modulator).

*GLM5: qualitative patterns of activations*. GLM5 included a regressor for all three incentives at two time points of interest: choice and rating moment, as well as a regressor at feedback moment. All regressors at the choice moment were modulated by a pmod of early certainty and button press (L/R). All regressors at the rating moment were modulated by a pmod of confidence judgment. The feedback regressor was modulated by accuracy. This GLM allowed us to investigate activity related to both baseline and the regression slope with early certainty or confidence judgment for these events.

*Regions of interest*. To avoid circular inference, we took an independent anatomical ROI of the VMPFC from the Brainnetome Atlas[71]. We included three areas along the ventral medial axis for the VMPFC ROI. Using this ROI, we extracted individual $t$-statistics (i.e., normalized beta estimates[50]) from contrasts of interest, and statistically compared them using paired $t$ tests or repeated-measure ANOVAs.

Moreover, in order to perform a finer-grained analysis into early certainty and confidence activations, we took a larger anatomical ROI, covering most of the medial prefrontal cortex from the Brainnetome Atlas[71] With this ROI, we extracted individual $t$-statistics from our contrasts of interest in GLM5 and averaged those activations over two dimensions (respectively, X and Z, and X and Y), so that we could assess the spread of activations over the last dimension, respectively, Y (anterior–posterior axis) and Z (ventral–dorsal axis).

**Reporting summary**. Further information on research design is available in the Nature Research Reporting Summary linked to this article.

## Data availability

All source data needed to evaluate or reproduce the figures and analyses described in the paper and supplementary materials are available online at 'https://doi.org/10.6084/m9.figshare.19228977'. Second-level neuroimaging maps can be found at 'https://neurovault.org/collections/12221/'[72].

## Code availability

All code needed to evaluate or reproduce the figures and analyses described in the paper and supplementary materials are available online at 'https://doi.org/10.6084/m9.figshare.19228977'.

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

## Acknowledgements

Data collection for this work was funded by two independent personal Amsterdam Brain and Cognition (ABC) Talent grants to J.L and R.v.H., and an NWO Veni Fellowship (grant 451-15-015) granted to M.L. M.L. is supported by a Swiss National Fund Ambizione Grant (PZ00P3_174127), J.L. is supported by an NWO VENI Fellowship grant (916-18-119).

## Author contributions

Conceptualization: R.J.v.H., J.L., M.L.; methodology: M.H., R.J.v.H., J.L., M.L.; data collection: M.H., N.s.d.B., G.B.; analyses: M.H., M.L.; writing original draft: M.H.; writing review & editing: M.H., G.B., N.s.d.B., A.G., D.D., R.vJ.H., J.L, M.L.; visualization: M.H., M.L.; Supervision: R.J.v.H., J.L., M.L.

## Competing interests

The authors declare no competing interests.
