## [Peer Review File · Communications Biology]

Reviewers' comments:

Reviewer #1 (Remarks to the Author):

Summary

In this paper, the authors attempt to illuminate how VMPFC activity is linked to incentives, expected value, and confidence judgments. Thirty-three subjects performed a 2-AFC contrast discrimination task where on each trial, Gabor patches were presented to the left and right of fixation for 150ms. Following presentation of the patches, subjects chose which patch had higher contrast, viewed the upcoming incentive, rated confidence in their perceptual decision, and received feedback. A thresholding procedure targeted 70% accuracy for each subject, although a large amount of variability in performance was evident (Figure 1B). Using a linear-mixed effects model with incentive net value and incentive absolute value as predictors, results revealed a positive effect of incentive net value on confidence, but no effect of incentive absolute value on confidence. The effect of incentive net value on confidence did not appear to be driven by any effect of incentive value on accuracy or reaction time (Figure 1C). Neuroimaging results showed that BOLD signal in the VMPFC correlates significantly with early certainty and incentives (Figure 3A/B), but VMPFC activity was only weakly associated with confidence judgments (Figure 3C). Other results were equivocal: while some analyses showed that VMPFC strongly correlates with "expected value" (an integration of confidence and incentive signals, Figure 3D), since VMPFC did not correlate with confidence in the loss context (Figure 5), this finding was not robust. The authors' final take-home message is that "although the VMPFC seems to encode both value and metacognitive signals, these metacognitive signals are only present during the prospect of gain and are disrupted in a context with loss or no monetary prospects."

Overall, additional details about the analyses and tasks need to be provided. Further, the exploratory nature of follow-up analyses and lack of a strong conclusion about the role of VMPFC in confidence diminishes the importance of the main findings. I outline major concerns and minor edits that are needed below.

Major Concerns

- In lines 138-139, you describe how the incentive-confidence interaction is not driven by any net incentive value effects on accuracy or reaction time (Figure 1C), but in lines 93-94 of the supplemental materials, you comment on how you found "a significant effect of RT on confidence, showing that quicker choices lead to higher confidence levels," as model #9, which includes Incentive and RT as predictors, was the winning model. Can you please further explain these differences, and why you conclude in the main manuscript that RT doesn't have an effect? Why is using RT as a predictor insignificant in your main manuscript's model, but significant in your winning model in the supplemental materials? Please also provide statistics in line 94 of the supplemental material on the "significant effect of RT on confidence."
- In your discussion section (lines 411-419), you state that your results have important implications, but you don't actually specify any implications for what you found informs any of the issues surrounding the role of VMPFC, VS, insula, and ACC in various psychological disorders. The takeaway message simply seems to be, "more research is needed." What are the "important implications" of this work? Please be more specific.
- You note that studies may need to target this problem with a finer spatial scale than univariate BOLD activations, but you make no attempt to use multivariate pattern analysis to see if more fine-grained patterns could be informative in this work.

Minor Edits/Comments

- Are the confidence judgments irrelevant for monetary reward in your behavioral task? And were subjects aware of this detail? Because I'm wondering how it may interact with incentives. For example, lines 435-436 state that participants earn "additional gains based on task performance." But if the incentives are tied to (Type 1) task performance alone, then is the confidence judgment irrelevant for monetary reward? Your description on lines 461-464 seems to indicate that this is the case, but I would like to confirm this detail.
- Line 137: The reference to Figure 1B here should be Figure 1C, yes?
- Also, the Figure 1C caption should provide some description of what "Inc." means. My

assumption is that Inc. = inclusive net value, and |Inc.| = Incentive absolute value, but this should be described.

- Line 141: Avoid the double-negative
- Line 437: Can you explain why one of your exclusion criteria is the "sufficient variation of confidence reports (std dev < 5 confidence points)"? If the thresholding procedure targets 70% performance, why would you expect confidence judgments to vary more than that?
- Line 453: Did you provide any other information to subjects about what it means to provide "accurate" confidence judgments in your instructions? For example, were they notified that if they select 50%, it's pure guessing, and 100%, they are "absolutely certain in their response?" If so, please describe it in the Methods.
- Line 389: "hence VMPFC confidence signal disrupted."
- How did you determine the sample size for this study?
- Figure 1 caption: "op 5%"

Reviewer #2 (Remarks to the Author):

The manuscript reported important empirical findings regarding the VMPFC's role in coding motivational and metacognitive information, and could potentially be a valuable addition to the literature in this field. The paper a clear conclusion that only in the gain condition was the VMPFC activity reliably correlated with confidence. However, to me, a series of analyses leading to this conclusion is quite hard to follow.

I understand the authors first tested a priori hypotheses, and then invented follow-up analyses after these hypotheses being rejected. Yet, the analyses shown in Figures 2-5 seem somewhat lacking consistency. Especially, the final conclusion was reached on the basis of the null results in the qualitative analysis, and these qualitative models were not directly compared with the models in Figure 2. The authors said "quantitative comparisons of hypotheses are notoriously hard to interpret", but I wonder why they have made this statement. Also, the figures only report standardized point estimates (t values) and did not convey goodness of fit indices, which makes it even harder to understand the reported results.

I wonder if it is possible to integrate the analyses shown in Figures 2-5 to make formal model comparisons of different GLM models. I suspect the analyses in Figure 5 can be done in concordance with those in Figure 3 by treating incentive as a categorical variable and evaluate its interaction with confidence to predict VMPFC activity.

The authors speculated that VMPFC confidence signal might be disrupted as the decision in the neutral condition is score-irrelevant. In relation to this, I suspect a possibility that confidence rating in the neutral condition (and possibly the loss condition) is noisier and less diagnostic of decision correctness than it is in the gain condition, which disrupts its correlation with DMPFC activity. I suggest evaluating metacognitive accuracy for each condition.

Also I would recommend calculating metacognitive accuracy for both confidence and early certainty. That could possibly shed some light on their correlation to the VMPFC activity.

Reviewers' comments:

Reviewer #1 (Remarks to the Author):

Summary

In this paper, the authors attempt to illuminate how VMPFC activity is linked to incentives, expected value, and confidence judgments. Thirty-three subjects performed a 2-AFC contrast discrimination task where on each trial, Gabor patches were presented to the left and right of fixation for 150ms. Following presentation of the patches, subjects chose which patch had higher contrast, viewed the upcoming incentive, rated confidence in their perceptual decision, and received feedback. A thresholding procedure targeted 70% accuracy for each subject, although a large amount of variability in performance was evident (Figure 1B). Using a linear-mixed effects model with incentive net value and incentive absolute value as predictors, results revealed a positive effect of incentive net value on confidence, but no effect of incentive absolute value on confidence. The effect of incentive net value on confidence did not appear to be driven by any effect of incentive value on accuracy or reaction time (Figure 1C).

Neuroimaging results showed that BOLD signal in the VMPFC correlates significantly with early certainty and incentives (Figure 3A/B), but VMPFC activity was only weakly associated with confidence judgments (Figure 3C). Other results were equivocal: while some analyses showed that VMPFC strongly correlates with “expected value” (an integration of confidence and incentive signals, Figure 3D), since VMPFC did not correlate with confidence in the loss context (Figure 5), this finding was not robust. The authors’ final take-home message is that “although the VMPFC seems to encode both value and metacognitive signals, these metacognitive signals are only present during the prospect of gain and are disrupted in a context with loss or no monetary prospects.”

Overall, additional details about the analyses and tasks need to be provided. Further, the exploratory nature of follow-up analyses and lack of a strong conclusion about the role of VMPFC in confidence diminishes the importance of the main findings. I outline major concerns and minor edits that are needed below.

Major Concerns

- In lines 138-139, you describe how the incentive-confidence interaction is not driven by any net incentive value effects on accuracy or reaction time (Figure 1C), but in lines 93-94 of the supplemental materials, you comment on how you found “a significant effect of RT on confidence, showing that quicker choices lead to higher confidence levels,” as model #9, which includes Incentive and RT as predictors, was the winning model. Can you please further explain these differences, and why you conclude in the main manuscript that RT doesn’t have an effect? Why is using RT as a predictor insignificant in your main manuscript’s model, but significant in your winning model in the supplemental materials? Please also provide statistics in line 94 of the supplemental material on the “significant effect of RT on confidence.”

We thank R1 for this question and apologize for the confusion.

Indeed, RT is a predictor of confidence but cannot (by design) be a predictor of the incentive bias on confidence. This is because incentives are revealed after the decision is made, hence after the RT is produced – thereby, RTs cannot integrate information about incentives.

In other words, consider the ideal scenario where a simple combination of two terms would perfectly explain confidence: decision uncertainty and incentive bias. Basically, RTs are

highly correlated with the share of confidence variance explained by decision uncertainty (hence with confidence, in general) but not with the share of confidence variance explained by the incentive bias.

In our methods section on lines 548-550 we state that for the behavioral analyses in the main text we modeled accuracy, reaction times and confidence as a function of absolute incentive value and net incentive value. Here we thus found that reaction times are not influenced by incentive value. In the supplementary materials we extended our original model of confidence to assess whether the effect of net incentive value on confidence would remain significant considering other factors that might influence confidence, among which reaction times. There we see that reaction times are negatively related to confidence, but our analyses in the main text clearly showed that there is no effect of net incentives on reaction times.

We now clarify this in the main text:

Lines 124-126: *‘Consequently, by design, there should not be any incentivization effects on either accuracy or reaction times as they develop during the choice.’*

Lines 145-148: *‘Next, to confirm the robustness of our main effect of net incentive on confidence, we ran several full linear mixed-effects models, which included additional control variables that could influence confidence as well (evidence, accuracy, reaction times, et cetera, see Supplementary Materials).’*

We also added to the supplementary materials on lines 93 – 95, the statistics regarding the incentive and RT effects on confidence, that can also be found in Table S3.

- In your discussion section (lines 411-419), you state that your results have important implications, but you don't actually specify any implications for what you found informs any of the issues surrounding the role of VMPFC, VS, insula, and ACC in various psychological disorders. The takeaway message simply seems to be, "more research is needed." What are the "important implications" of this work? Please be more specific.

We thank R1 for this important question.

We believe that the knowledge on motivation, confidence and their neurobiological processes from this work on healthy individuals constitutes a first stepping stone and has some important implications for studying populations where these processes go awry. It is crucial for our behavioral control and adaptation that our confidence is in line with reality. Discrepancies between behavior and confidence in that behavior have been described in various psychiatric disorders and could give rise to pathological decision-making (Hoven et al., 2019).

Our results replicate the incentive bias on confidence and build upon the literature showing that affective and motivational states can influence confidence. Research has shown that patients suffering from various psychiatric disorders, such as obsessive-compulsive disorder, addiction or schizophrenia, have deficits in reward and motivational processes, that are accompanied by dysregulated neural circuitries (Admon et al., 2012; Choi et al., 2012; Clark et al., 2019; Koob and Volkow, 2016; Strauss et al., 2014). These deficits in motivational processes could have an effect on confidence judgments as well since we showed that motivational states can bias confidence. It is important to study how confidence might

contribute to the symptomatology of these psychiatric disorders: could there be an interaction between motivation and confidence that could fuel dysfunctional behaviors?

Furthermore, this paper showed that the VMPFC is a key brain region involved in this interaction between incentive motivation and metacognition. This has implications for hypotheses on the neurobiological basis of confidence abnormalities in psychiatry. The VMPFC has a pivotal role in multiple aspects of mental health, and its function is affected in multiple psychiatric disorders (Chai et al., 2011; Goldstein and Volkow, 2011; Hiser and Koenigs, 2018; Thorsen et al., 2018). Assessing the (dys)function of the VMPFC could also impact clinical practice, as it could help with treating and prediction responses in mental illnesses (Hiser and Koenigs, 2018). The current study thus gives rise to studying neurobiological explanations for confidence abnormalities and their interaction with incentive motivation in clinical populations.

We have now addressed this more specifically in the Discussion on lines 423-440:

“Our results constitute a first stepping stone and have important implications for studying clinical populations where these (meta)cognitive processes go awry. It shows that motivational processes can influence confidence and when there are discrepancies between one’s behavior and confidence in that behavior this could give rise to pathological decision making. Indeed, several psychiatric disorders such as addiction, obsessive-compulsive disorder and schizophrenia have been associated with disrupted incentive processing (Admon et al., 2012; Choi et al., 2012, Clark et al., 2019, Koob & Volkow, 2016; Strauss et al., 2014), and studies have additionally demonstrated distorted confidence estimations in these groups (Hoven et al., 2019). Our study indicates that the VMPFC is a key region involved in the interaction between motivation and metacognition. VMPFC function is also often affected in many psychiatric disorders (Hiser & Koenigs, 2018). The current study provides a means of studying neurobiological explanations for confidence abnormalities and their interaction with incentive motivation in clinical populations which can potentially impact clinical practice, as it could help treat psychopathology (Hiser & Koenigs, 2018). Therefore, the relation between motivational processes and confidence estimation and their role in psychopathology warrants future investigation.”

- You note that studies may need to target this problem with a finer spatial scale than univariate BOLD activations, but you make no attempt to use multivariate pattern analysis to see if more fine-grained patterns could be informative in this work.

We thank R1 for making this important point, which is something we indeed have considered. After carefully considering the methods and considering multiple concerns with MVPA, we finally decided not to use this analysis here. First, note that we see robust univariate signals for the early certainty (all conditions) and for confidence in the gain condition. Why confidence in the loss condition should specifically be encoded in a multivariate rather than univariate code seems quite improbable. Yet, we did include the analyses reported in Figure 6, where the VMPFC activations were assessed across the anterior-posterior and ventro-dorsal gradients in the current paper. The results from these analyses also seem to indicate that finer-grained information would not change our conclusions. The reasons for our decision to not include additional MVPA analyses are the following:

A recent paper on best practices for MVPA analyses by (Poldrack et al., 2020) describes that having an adequate sample size is essential for the accuracy of the prediction analyses. The authors state that: ‘*predictive analyses should not be performed with samples smaller than several **hundred** observations*’, and that using small samples can lead to ‘*highly variable estimates of predictive accuracy*’, which together with publication biases have led to ‘*a body of literature with inflated estimates of predictive accuracy*’. This view is supported by a paper by (Varoquaux, 2018), which indeed showed that sample sizes typical for fMRI experiments (including ~30 subjects, like our study) inherently lead to large error bars, compromising the reliability of conclusions that can be drawn from them.

Another recent study also called for caution in the interpretation of MVPA results, stating that fMRI signals are limited for studying ‘*coordinated coding across voxels*’, concluding that ‘*care should be taken in interpreting significant MVPA results as representing anything beyond a collection of univariate effects*’ (Pakravan and Ghazizadeh, 2021).

Therefore, while we agree that MVPA is an intriguing method that can be very informative, considering these cautions and limitations we were not convinced that it would be extra informative in answering our research questions.

Minor Edits/Comments

- Are the confidence judgments irrelevant for monetary reward in your behavioral task? And were subjects aware of this detail? Because I'm wondering how it may interact with incentives. For example, lines 435-436 state that participants earn "additional gains based on task performance." But if the incentives are tied to (Type 1) task performance alone, then is the confidence judgment irrelevant for monetary reward? Your description on lines 461-464 seems to indicate that this is the case, but I would like to confirm this detail.

We thank R1 for this comment, which is a very relevant question. We used a version of our task where monetary reward was based on performance and not on confidence judgments. We previously showed that this performance-based version of the task elicits a similar incentive bias as when reward was based on the accuracy of confidence judgements (Lebreton et al., 2018), while keeping confidence sensitivity identical across conditions (see **Rebuttal Figure 1**).

Rebuttal Figure 1 – from (Lebreton et al., 2018).

We clarified this design choice in the main text on lines 475-483:

“Subjects were instructed that they would only be rewarded based on their performance (i.e. they should be as accurate as possible to maximize their earnings), and that it was important to give accurate confidence judgments. (...) Thus, performance but not confidence was

incentivized. According to our previous findings (Lebreton et al., 2018) this design elicits incentive bias on confidence while keeping confidence sensitivity identical across conditions – an important consideration when interpreting differences in confidence activations between those conditions”

- Line 137: The reference to Figure 1B here should be Figure 1C, yes?

We thank R1 for noticing this error. We have adjusted it in the main text.

- Also, the Figure 1C caption should provide some description of what "Inc." means. My assumption is that Inc. = inclusive net value, and |Inc.| = Incentive absolute value, but this should be described.

We thank R1 for noticing this. We have added this to the caption of Figure 1.

- Line 141: Avoid the double-negative

We thank R1 for this comment. We have adjusted it in the main text: *‘Moreover, we did not find evidence for an effect of absolute incentive value on both accuracy and RT’.*

- Line 437: Can you explain why one of your exclusion criteria is the "sufficient variation of confidence reports (std dev < 5 confidence points)"? If the thresholding procedure targets 70% performance, why would you expect confidence judgments to vary more than that?

We thank R1 for this question and apologize for the confusion. The thresholding/calibration procedure does not specifically target 70% performance, but estimates the full psychometric function, that we use to elicit different difficulty levels. During the calibration, the distribution of contrast difference between the Gabor patches (i.e. difficulty) was adapted using a staircase procedure to reach ~70% performance. The calibration data were then used to estimate the psychometric function for each individual. The estimated parameters of the psychometric function were applied to generate stimuli for the confidence task, spanning three defined difficulty levels for all incentive conditions (i.e. targeting 65%, 75% and 85% accurate responses, on average). This should result in more variation in performance and thus also in confidence reports, which is why we set this exclusion criterium.

We have clarified this in the text:

Lines 485-487: *‘This was done using a staircase procedure, which data were used to estimate a full psychometric function, whose parameters were used to generate stimuli for the main task, spanning three difficulty levels (i.e. 65%, 75% and 85% accuracy on average).’*

- Line 453: Did you provide any other information to subjects about what it means to provide "accurate" confidence judgments in your instructions? For example, were they notified that if they select 50%, it's pure guessing, and 100%, they are "absolutely certain in their response?" If so, please describe it in the Methods.

We thank R1 for this comment and apologize for the confusion.

Indeed, we provided the subjects with additional information as to what it meant to provide accurate confidence judgments. We explained to them that 50% means a guess, and 100% that they are absolutely certain that they made the correct choice.

We have now described this in the Methods:

Lines 478-479: *'They were notified that 50% confidence would signal that they made a guess, whereas 100% confidence would signal that they were absolutely certain that they made the correct choice.'*

- Line 389: "hence VMPFC confidence signal disrupted."

We thank R1 for this comment, and we changed this in the text:

'which could result in a disrupted VMPFC confidence signal

- How did you determine the sample size for this study?

We thank R1 for this important question.

We did not perform an a-priori power analysis for this study to determine the sample size. We based our sample size on sample sizes that are customary in the field, such as those used in other research studying the behavioral interaction between confidence and incentives (Lebreton et al., 2018, 2019; Ting et al., 2020) and fMRI studies into confidence signals (Morales et al., 2018; Rouault and Fleming, 2020; Rouault et al., 2021).

We performed an ex-post sample size calculation using GPower, based on a quick meta-analysis of four previous fMRI datasets studying confidence/value (Lebreton et al., 2009, 2012, 2013, 2015). This revealed that, in an independent VMPFC region of interest, the estimated Cohen's d for a random-effect analysis on individual's "values" and/or "confidence" is $d \sim 1$. This means that, using a similar scanning protocol and statistical models, a sample size of $N = 28$ is required to reach a power of 95% with an α -rate of 0.001 (which is the classical voxel-level threshold used to generate cluster-size corrections for multiple comparisons at the whole brain level).

Because those analyses are ex-post, we did not report them in the manuscript.

- Figure 1 caption: "op 5%"

We thank R1 for their comment, and have changed this in the caption to: *'of 5%'*

Reviewer #2 (Remarks to the Author):

The manuscript reported important empirical findings regarding the VMPFC's role in coding motivational and metacognitive information, and could potentially be a valuable addition to the literature in this field. The paper a clear conclusion that only in the gain condition was the VMPFC activity reliably correlated with confidence. However, to me, a series of analyses leading to this conclusion is quite hard to follow.

I understand the authors first tested a priori hypotheses, and then invented follow-up analyses after these hypotheses being rejected. Yet, the analyses shown in Figures 2-5 seem somewhat lacking consistency. Especially, the final conclusion was reached on the basis of the null results in the qualitative analysis, and these qualitative models were not directly compared with the models in Figure 2. The authors said "quantitative comparisons of hypotheses are notoriously hard to interpret", but I wonder why they have made this statement. Also, the figures only report standardized point estimates (t values) and did not convey goodness of fit indices, which makes it even harder to understand the reported results.

We thank R2 for this comment, which allows us to clarify our reasoning.

First, note that all the models evaluated in Figures 2-5 are *de facto* nested in the qualitative model comparison exercise (only a subset of them are explicitly graphically represented in Figure 5A). Second, the results of the qualitative analyses are not null: we find very significant effects of incentives at baseline, of early certainty in all conditions, and of confidence in gain condition. This rules out the possibility that the lack of effect of confidence in the neutral and loss condition are due to a lack of power. Instead, we claim that we have identified a significant pattern that is different from the ones that we could imagine ex-ante, informed by the previous literature (Figure 2-5).

Then, as correctly pointed out by R2, we state (along with other authors see e.g. (Palminteri et al., 2017; Pitt and Myung, 2002; Roberts and Pashler, 2000)) that "*Quantitative comparisons of hypotheses are notoriously hard to interpret*". Indeed, although model comparison procedures will always find a (relatively) better model in a model space, nothing guarantees that it is *a good model* (Pitt and Myung, 2002; Roberts and Pashler, 2000). Here, by dissecting the actual patterns of correlation across the different conditions, we can show that what seems to be the "Best Model" (the Expected Value model) is actually not good enough, as it does not account for the pattern of activations actually observed. In other terms, the model is *falsified* by specific patterns in the data (absence of correlation with confidence in the loss condition) (Palminteri et al., 2017).

Note that we initially performed proper Bayesian Model Selection (BMS) in our original model space (GLM1, 2a, 2b, 3 and 4). BMS was performed using the MACS (Model Assessment, Comparison and Selection) toolbox for SPM12 (Soch & Allefeld, 2018). Random effects (RFX) BMS estimates how frequently each model is optimal in all voxels (either whole-brain or all voxels in a given ROI) of all subjects, and gives rise to likeliest frequencies (LF) (i.e. the posterior modes) and exceedance probabilities (EP). LF can be interpreted as the proportion of subjects in which a particular GLM is optimal, and EP is the posterior probability that a given model is more frequently optimal than all other models in the model space. Thus, the optimal model is the one that best explains the signal in most

voxels, and therefore the one with the largest LF and EP. Consequently we use these two quantities when we make quantitative statements about model selection.

We compared GLM1, 2a, 2b, 3 and 4 to explore how well these models explained activity patterns in our ROI of the vmPFC. For each subject and model, cross-validated log model evidence (cvLME) maps were estimated, indicating the model’s performance for each voxel in the brain. Finally, we performed a RFX BMS, which accounts for the optimal model to vary across voxels and subjects.

This procedure came up quite inconclusive. Yet, while no model clearly wins the model-comparison, the relative pattern of EP and LF surprisingly revealed that the model with incentive and early certainty at rating moment (GLM2b) had the highest exceedance probability (EP), as well as likeliest frequency (LF) within our ROI of the vmPFC – a finding in apparent contradiction with our regressor comparison exercise (which favors GLM3) as well as with our qualitative pattern analysis. Intrigued by this results, we replicated this procedure by selectively removing one model of the model-space – to our surprise this completely changed the relative EP and LF of the remaining models (**Rebuttal Figure 2**). For instance, removing GLM4 (with only incentives at the rating period) from the model space gives GLM3 (expected value) a relative advantage over GLM2b. Therefore, we finally chose to not rely on this procedure, which output seems inconclusive and/or difficult to interpret in the present case.

Rebuttal Figure 2. Results on a BMS in the VMPFC ROI

Finally, most of our figures indeed report tests on fMRI individual regressors (t-values). Testing and comparing these point estimates, as proxy for ‘brain activations’, has been the dominant class of inference in the field of functional neuroimaging for decades. We are not aware of goodness-of-fit measures that would be routinely computed by the dominant neuroimaging analyses software (SPM, FSL, etc.) and which would have been properly evaluated to draw inferences in fMRI.

I wonder if it is possible to integrate the analyses shown in Figures 2-5 to make formal model comparisons of different GLM models. I suspect the analyses in Figure 5 can be done in concordance with those in Figure 3 by treating incentive as a categorical variable and evaluate its interaction with confidence to predict VMPFC activity.

Although it is possible to design a model tailored to account for the observed pattern of data (i.e. featuring specific correlations with confidence in the gain domain), we are reluctant to include it in a proper, quantitative model comparison with our other models, as suggested by R2. The reason is that this model would be designed after seeing the data (i.e. *ex-post*). In contrast, the other models have been designed based purely on theory (i.e. *ex ante*, as theoretical hypotheses).

Thereby, in the highly likely case the model comparison would identify this new model as the best, it would have benefitted from an unfair advantage – actually, given that models are quantitative hypotheses, this is analogous to ‘HARKING’, i.e., designing Hypotheses After the Results are Known (Kerr, 1998).

In the improbable case the model comparison would not identify this model as the best, the result would be very hard to interpret (see previous point). Therefore, we argue that the current, chronological narrative of our analyses gives a better account of the relative merits of the different theoretical accounts of our data.

The authors speculated that VMPFC confidence signal might be disrupted as the decision in the neutral condition is score-irrelevant. In relation to this, I suspect a possibility that confidence rating in the neutral condition (and possibly the loss condition) is noisier and less diagnostic of decision correctness than it is in the gain condition, which disrupts its correlation with DMPFC activity. I suggest evaluating metacognitive accuracy for each condition.

Also I would recommend calculating metacognitive accuracy for both confidence and early certainty. That could possibly shed some light on their correlation to the VMPFC activity.

This is an excellent point, that indeed deserves a more thorough treatment.

As also mentioned in response to one of R1's comments, we used a version of our task where monetary reward (incentivization) was based on choice performance and not on confidence judgments. We previously showed that this performance-based version of the task elicits a similar incentive bias as when reward was based on the accuracy of confidence judgements (Lebreton et al., 2018), while keeping confidence sensitivity identical across conditions (Rebuttal Figure 3).

Rebuttal Figure 3 – from (Lebreton et al., 2018).

We clarified this design choice in main text on lines 475 – 483:

“Subjects were instructed that they would only be rewarded based on their performance (i.e. they should be as accurate as possible to maximize their earnings), and that it was important to give accurate confidence judgments. (...) Thus, performance but not confidence was incentivized. According to our previous findings (Lebreton et al., 2018) this design elicits incentive bias on confidence while keeping confidence sensitivity identical across conditions – an important consideration when interpreting differences in confidence activations between those conditions”.

Note that several indexes measure metacognitive sensitivity: from simple ones like discrimination (confidence_{correct} – confidence_{incorrect}) to meta-d' that can be computed using standard or hierarchical Bayesian procedures. Practically, these latter ones are respectively implemented in the MATLAB code of (Maniscalco and Lau, 2012) available at www.columbia.edu/~bsm2105/type2sdt/, and in the HMeta-d package described in (Fleming, 2017) available <https://github.com/smfleming/HMeta-d>. In the 5 datasets that constituted our previous paper (Lebreton et al., 2018), discrimination and meta-d' (estimated with the standard procedure) were highly correlated (**Rebuttal Figure 4**), and provided identical conclusions with respect to the effects of incentives on confidence judgments.

Rebuttal Figure 4. Correlations between meta-d' and discrimination. The 4 experiments are from (Lebreton et al., 2018). Each dot represent an incentive condition in one subject. Meta-d' was evaluated with the MLE package from (Maniscalco and Lau, 2012).

We nonetheless checked that this findings replicates in the current manuscript's data. We therefore evaluated metacognitive sensitivity in our different conditions, using the different indexes at our disposal. First, we estimated the correlations between the different measures of metacognitive sensitivity (**Rebuttal Figure 5**).

Rebuttal Figure 5. Correlations between sensitivity indexes in the present data. Each dot represent an incentive condition in one subject. Meta-d' (MLE) was evaluated with the MLE package from (Maniscalco and Lau, 2012). Meta-d' (HB) was evaluated with the hierarchical-Bayesian package from (Fleming, 2017).

This analysis revealed that meta-cognitive sensitivity is more noisy in the present data, notably due to lower number of trial per condition ($n = 48$). Because meta-d' computations are known to be imprecise in those low-power designs (Rouault et al., 2018), and because discrimination was the index that showed the highest correlations with the other two, we propose to report results from discrimination analyses.

Replicating our (Lebreton et al., 2018) findings, we found that the incentive condition did not have a significant effect on metacognitive sensitivity in this performance-incentivized version of our task: ($F(2,62) = 0.25, p = 0.783$).

We now report this in the main text:

Results section on lines 150-154:

‘Lastly, we tested for an incentive effect on metacognitive sensitivity – a metric that measures the efficacy with which subjects discriminate between correct and incorrect answers using their confidence ratings. Replicating earlier findings (Lebreton et al., 2018), we found that incentive condition did not have a significant effect on metacognitive sensitivity ($F(2,62) = 0.25, p = 0.783$. Loss: 5.5973 ± 1.2106 , neutral: 4.8572 ± 1.0515 , gain: 5.2797 ± 0.8692).’

Method section on lines 532-541:

‘Metacognitive sensitivity:

Metacognitive sensitivity is a metric that indicates how well an observer's confidence judgments discriminate between their correct and incorrect answers and can be represented using several indexes. For example, discrimination is a metric calculated as the difference between the average confidence for correct answers and the average confidence for incorrect answers, whereas meta-d' is a metric based on the Signal Detection Theory (SDT) framework. Notably, meta-d' computations are known to be imprecise in designs with low number of trials per condition (Rouault et al., 2018). This, together with results from our earlier work (Lebreton et al., 2018) showing high correlations between discrimination and meta-d', as well as identical conclusions with respect to the effects of incentives on these measures, we used the discrimination metric as our measure of metacognitive sensitivity.’

Discussion section on lines 551-553:

'For the analysis of metacognitive sensitivity, we performed a repeated measures ANOVA, with net incentive value as within-subject factor.'

Literature

- Admon, R., Bleich-Cohen, M., Weizmant, R., Poyurovsky, M., Faragian, S., and Hendler, T. (2012). Functional and structural neural indices of risk aversion in obsessive-compulsive disorder (OCD). *Psychiatry Res. - Neuroimaging* 203, 207–213.
- Chai, X.J., Whitfield-Gabrieli, S., Shinn, A.K., Gabrieli, J.D.E., Nieto Castañón, A., McCarthy, J.M., Cohen, B.M., and Öngür, D. (2011). Abnormal Medial Prefrontal Cortex Resting-State Connectivity in Bipolar Disorder and Schizophrenia. *Neuropsychopharmacology* 36, 2009–2017.
- Choi, J., Shin, Y., Jung, W.H., Jang, J.H., and Kang, D. (2012). Altered Brain Activity during Reward Anticipation in Pathological Gambling and Obsessive-Compulsive Disorder. *PLoS ONE* 7, 3–10.
- Clark, L., Boileau, I., and Zack, M. (2019). Neuroimaging of reward mechanisms in Gambling disorder: an integrative review. *Mol. Psychiatry* 24, 674–693.
- Fleming, S.M. (2017). HMeta-d: hierarchical Bayesian estimation of metacognitive efficiency from confidence ratings. *Neurosci. Conscious.* 2017.
- Goldstein, R.Z., and Volkow, N.D. (2011). Dysfunction of the prefrontal cortex in addiction: neuroimaging findings and clinical implications. *Nat. Rev. Neurosci.* 12, 652–669.
- Hiser, J., and Koenigs, M. (2018). The Multifaceted Role of the Ventromedial Prefrontal Cortex in Emotion, Decision Making, Social Cognition, and Psychopathology. *Biol. Psychiatry* 83, 638–647.
- Hoven, M., Lebreton, M., Engelmann, J.B., Denys, D., Luigjes, J., and van Holst, R.J. (2019). Abnormalities of confidence in psychiatry: an overview and future perspectives. *Transl. Psychiatry* 9, 1–18.
- Kerr, N.L. (1998). HARKing: Hypothesizing After the Results are Known. *Personal. Soc. Psychol. Rev.* 2, 196–217.
- Koob, G.F., and Volkow, N.D. (2016). Neurobiology of addiction: a neurocircuitry analysis. *Lancet Psychiatry* 3, 760–773.
- Lebreton, M., Jorge, S., Michel, V., Thirion, B., and Pessiglione, M. (2009). An Automatic Valuation System in the Human Brain: Evidence from Functional Neuroimaging. *Neuron* 64, 431–439.
- Lebreton, M., Kawa, S., d’Arc, B.F., Daunizeau, J., and Pessiglione, M. (2012). Your Goal Is Mine: Unraveling Mimetic Desires in the Human Brain. *J. Neurosci.* 32, 7146–7157.
- Lebreton, M., Bertoux, M., Boutet, C., Lehericy, S., Dubois, B., Fossati, P., and Pessiglione, M. (2013). A Critical Role for the Hippocampus in the Valuation of Imagined Outcomes. *PLOS Biol.* 11, e1001684.
- Lebreton, M., Abitbol, R., Daunizeau, J., and Pessiglione, M. (2015). Automatic integration of confidence in the brain valuation signal. *Nat. Neurosci.* 18, 1159–1167.
- Lebreton, M., Langdon, S., Sliker, M.J., Nooitgedacht, J.S., Goudriaan, A.E., Denys, D., van Holst, R.J., and Luigjes, J. (2018). Two sides of the same coin: Monetary incentives concurrently improve and bias confidence judgments. *Sci. Adv.* 4, eaaq0668.
- Lebreton, M., Bacily, K., Palminteri, S., and Engelmann, J.B. (2019). Contextual influence on confidence judgments in human reinforcement learning. *PLOS Comput. Biol.* 15, e1006973.

- Maniscalco, B., and Lau, H. (2012). A signal detection theoretic approach for estimating metacognitive sensitivity from confidence ratings. *Conscious. Cogn.* *21*, 422–430.
- Morales, J., Lau, H., and Fleming, S.M. (2018). Domain-General and Domain-Specific Patterns of Activity Supporting Metacognition in Human Prefrontal Cortex. *J. Neurosci.* *38*, 3534–3546.
- Pakravan, M., and Ghazizadeh, A. (2021). Coordinated multivoxel coding beyond univariate effects is not likely to be observable in fMRI data. *BioRxiv* 2021.06.13.448229.
- Palminteri, S., Wyart, V., and Koechlin, E. (2017). The Importance of Falsification in Computational Cognitive Modeling. *Trends Cogn. Sci.* *21*, 425–433.
- Pitt, M.A., and Myung, I.J. (2002). When a good fit can be bad. *Trends Cogn. Sci.* *6*, 421–425.
- Poldrack, R.A., Huckins, G., and Varoquaux, G. (2020). Establishment of Best Practices for Evidence for Prediction: A Review. *JAMA Psychiatry* *77*, 534–540.
- Roberts, S., and Pashler, H. (2000). How persuasive is a good fit? A comment on theory testing. *Psychol. Rev.* *107*, 358–367.
- Rouault, M., and Fleming, S.M. (2020). Formation of global self-beliefs in the human brain. *Proc. Natl. Acad. Sci.* *117*, 27268–27276.
- Rouault, M., McWilliams, A., Allen, M.G., and Fleming, S.M. (2018). Human metacognition across domains: insights from individual differences and neuroimaging. *Personal. Neurosci.* *1*.
- Rouault, M., Lebreton, M., and Pessiglione, M. (2021). A shared brain system forming confidence judgment across cognitive domains.
- Soch, J., & Allefeld, C. (2018). MACS – a new SPM toolbox for model assessment, comparison and selection. *Journal of Neuroscience Methods*, *306*, 19–31.
- Strauss, G.P., Waltz, J.A., and Gold, J.M. (2014). A Review of Reward Processing and Motivational Impairment in Schizophrenia. *Schizophr. Bull.* *40*, S107–S116.
- Thorsen, A.L., Hagland, P., Radua, J., Mataix-Cols, D., Kvale, G., Hansen, B., and van den Heuvel, O.A. (2018). Emotional Processing in Obsessive-Compulsive Disorder: A Systematic Review and Meta-analysis of 25 Functional Neuroimaging Studies. *Biol. Psychiatry Cogn. Neurosci. Neuroimaging* *3*, 563–571.
- Ting, C.-C., Palminteri, S., Engelmann, J.B., and Lebreton, M. (2020). Robust valence-induced biases on motor response and confidence in human reinforcement learning. *Cogn. Affect. Behav. Neurosci.*
- Varoquaux, G. (2018). Cross-validation failure: Small sample sizes lead to large error bars. *NeuroImage* *180*, 68–77.

Reviewers' comments:

Reviewer #1 (Remarks to the Author):

Summary

In this revised manuscript, the authors addressed my criticisms from the first submission. Overall, I am satisfied with their replies, but I have a few remaining concerns and comments that I would like to see addressed in a revision.

Major Concerns

- I have two additional questions about Figure 3C, which may generalize to other analyses: you mention that activations shown here are “mirrored in the negative correlation with confidence, suggesting these brain regions are part of the visuo-motor network that processes the movement of the cursor on the rating scale.” My first question is: can you distinguish between the motor-related activity and the regions computing confidence? If so, can that be shown more clearly? For example, one of the papers you cite (Morales et al., 2018) creates a mask for regions involved in motor activity, to leave them out of analyses related to confidence. However, in your Figure 3C, both motor-related activations and confidence-related activations are all lumped together. My second question is: can you comment on whether motor-related activity is a part of the activity shown in Figure 3A and 3B? I think you have controlled for it in this case (re: line 601), but I want to be sure. Motor-related activity is a confound that should be addressed before the paper is published.

Minor Edits

- I still find the article somewhat difficult to read, as terms are not well-defined or explained in advance of their usage. For example, what is the difference between the “incentive net value” (line 137) and the “incentive condition” (line 152)? Based on the Figure 1B Caption, it appears these are synonyms. Why are you introducing two terms to refer to the same concept? Can you please go through your manuscript and at the first use of each term, verify whether it has been previously defined?

- Line 149 – You mention that you test for an incentive effect on metacognitive sensitivity, but you don't provide a citation or definition of the exact measure you're using. Is this meta-d' from Maniscalco & Lau 2012? Or something else?

Minor Comments

- For what it's worth, I don't agree that Poldrack's and Varoquaux's comments about best practices for using MVPA in fMRI should be justification to avoid using MVPA to study prefrontal areas, where neural coding is complex and marked by mixed selectivity (e.g., Fusi, Miller, and Rigotti, 2016). MVPA methods can reveal important insights about metacognition in prefrontal areas, some of which you cite in this manuscript (e.g., Morales et al., 2018). So while I do not doubt that having sample sizes of hundreds of individuals increases the robustness of results, I do not think these methods should be avoided in paradigms with fewer subjects. (e.g., look at Huth et al., Nature, 2016. Plenty of interesting insights from using machine learning analyses with small N, with proper division of training/test sets, etc.).

Reviewer #2 (Remarks to the Author):

The authors have successfully addressed my concerns and now I recommend publication of the manuscript.

Reviewers' comments:

Reviewer #1 (Remarks to the Author):

Summary

In this revised manuscript, the authors addressed my criticisms from the first submission. Overall, I am satisfied with their replies, but I have a few remaining concerns and comments that I would like to see addressed in a revision.

We thank R1 for their appreciation of our first revision. We hope that this new set of revisions will satisfactorily address all their remaining concerns.

Major Concerns

- I have two additional questions about Figure 3C, which may generalize to other analyses: you mention that activations shown here are “mirrored in the negative correlation with confidence, suggesting these brain regions are part of the visuo-motor network that processes the movement of the cursor on the rating scale.” My first question is: can you distinguish between the motor-related activity and the regions computing confidence? If so, can that be shown more clearly? For example, one of the papers you cite (Morales et al., 2018) creates a mask for regions involved in motor activity, to leave them out of analyses related to confidence. However, in your Figure 3C, both motor-related activations and confidence-related activations are all lumped together. My second question is: can you comment on whether motor-related activity is a part of the activity shown in Figure 3A and 3B? I think you have controlled for it in this case (re: line 601), but I want to be sure. Motor-related activity is a confound that should be addressed before the paper is published.

We thank R1 for this remark and their questions. Regarding the first question, and following R1’s suggestion, we now use a mask for motor regions, so that we can more clearly tease apart the confidence-related activity from the motor-related activity, inspired by Morales et al., (2018). To do so, we leverage ‘Neurosynth’, which is a platform for large-scale, automated synthesis of functional magnetic resonance imaging (fMRI) data, using the term ‘motor’. In the present case, the mask was generated using a meta-analysis of 2565 studies, that includes key regions with motor-related activity (<https://neurosynth.org/analyses/terms/motor/>).

We used this mask as an exclusive mask, so that the remaining activity patterns we observe cannot be attributed to motor activities. On lines (612-617), in the method section we have added the following paragraph:

‘Importantly, to ensure that our brain activations of interest (i.e. related to early certainty, incentive and confidence) were not confounded by motor-related activations, we performed control analyses that implemented an exclusive masking for motor activations. To do so, we generated the exclusive mask from ‘Neurosynth’ (a platform for large-scale, automated synthesis of fMRI data (Yarkoni et al., 2011)), using the term ‘motor’ (<https://neurosynth.org/analyses/terms/motor/>). This mask represents key regions related to motor processes as identified by an automated meta-analysis of 2565 studies.’

On lines (205-211), in the results section we have added the following:

‘As observed with confidence activations, motor-related activity can be an important confound. To ensure that our activity patterns of interest (i.e. early certainty, incentive and confidence) were not related to motor processes, we replicate our analyses using an exclusive motor-related mask, generated from large-scale automated meta-analyses (see Methods for more details). Importantly, those control analyses revealed that most activations – with the exception of the visuo-motor

activations identified in the confidence activation maps – remain significantly associated to our variables of interest (for whole-brain activation tables when using this exclusive mask, see Supplementary Materials).’

Regarding the second question of whether motor-related activity is a part of the activity shown in Figure 3A and 3B, we also ran the above-mentioned analyses, using the exclusive mask related to motor processes, for early certainty processing (Figure 3A) and incentive processing (Figure 3B). The full activation tables reporting the masked activations are available in the Supplementary Materials.

We thank the reviewer for this comment and are confident we now addressed motor confounds in these analyses, as the reviewer suggested. We were not too concerned that our VMPFC findings were caused by motor confounds to begin with – although we completely endorse controlling for motor confounds in fMRI studies – since participants had to press the same buttons and perform the same type of motor actions in all three incentive conditions. Because of this, it is unlikely that our main findings of distinct VMPFC confidence signals in the various incentive conditions was caused by motor effects.

Minor Edits

- I still find the article somewhat difficult to read, as terms are not well-defined or explained in advance of their usage. For example, what is the difference between the “incentive net value” (line 137) and the “incentive condition” (line 152)? Based on the Figure 1B Caption, it appears these are synonyms. Why are you introducing two terms to refer to the same concept? Can you please go through your manuscript and at the first use of each term, verify whether it has been previously defined?

We thank R1 for this helpful comment, which will improve the readability of the article.

Incentive condition refers to the key experimental manipulation that we implemented, and that define gains, neutral and loss trials. Incentive net value and incentive absolute value refer to *variables* that are we use in our analysis of the effects of this manipulation.

We now try to clarify this terminology both at the beginning of the result section:

Lines 116-117: *‘Then, we experimentally manipulated the available monetary outcomes, defining several incentive conditions: at each trial, participants could win (gain context) or lose (loss context) points – or not gain or lose anything (neutral context) – depending on the correctness of their choice.’*

Lines 133-139: *‘First, using an approach similar to Lebreton et al. (2018), we used linear mixed-effect models to evaluate the effects of our experimental manipulation of incentives (i.e. the incentive condition) on behavioral variables (see Methods). More specifically, we defined and tested the incentives’ biasing effects (i.e. the net incentive value, or in other words, the linear effect of incentives coded as -1, 0 and +1) and incentives’ motivational effects (i.e. the absolute incentive value, or in other words, the mere presence of incentives, indicating whether something is at stake coded as 0 and +1).*

And in the methods on lines 546-550: *‘In order to analyze the effect of the incentive condition (i.e. of our experimental manipulation of incentives), for all three trial-by-trial dependent variables we used the absolute incentive value (i.e. the absolute value of the monetary incentive, $|V|$), coded as 0 and +1) and the net incentive value (i.e. the linear value of the monetary incentive, V , coded as -1, 0 and +1) as predictor variables.’*

- Line 149 – You mention that you test for an incentive effect on metacognitive sensitivity, but you don't provide a citation or definition of the exact measure you're using. Is this meta-d' from Maniscalco & Lau 2012? Or something else?

We thank R1 for pointing out that this was not clearly mentioned in the results. We defined the measure for metacognitive sensitivity in the methods, and now refer to this on line 154: *'Lastly, we tested for an incentive effect on metacognitive sensitivity – a metric that measures the efficacy with which subjects discriminate between correct and incorrect answers using their confidence ratings (see Methods for details on its' computation).'*

Minor Comments

- For what it's worth, I don't agree that Poldrack's and Varoquaux's comments about best practices for using MVPA in fMRI should be justification to avoid using MVPA to study prefrontal areas, where neural coding is complex and marked by mixed selectivity (e.g., Fusi, Miller, and Rigotti, 2016). MVPA methods can reveal important insights about metacognition in prefrontal areas, some of which you cite in this manuscript (e.g., Morales et al., 2018). So while I do not doubt that having sample sizes of hundreds of individuals increases the robustness of results, I do not think these methods should be avoided in paradigms with fewer subjects. (e.g., look at Huth et al., Nature, 2016. Plenty of interesting insights from using machine learning analyses with small N, with proper division of training/test sets, etc.).

We thank R1 for their comment. We absolutely agree that MVPA methods can reveal important insights and very relevant new findings, and is an interesting avenue of research when one has the appropriate number of observations. While we agree that increasing the number of subjects in a sample size would improve the robustness of results, our main concern with using MVPA analyses in the current article was not just the low sample size per se, but more specifically the low number of trials per condition (low number of observations per condition per subject). Note that indeed, in our case, participants only perform 24 trials per session in each incentive condition. Therefore, all generalization tests based on MVPA that could be interesting for the present article (e.g. training on gain condition and testing on loss condition) would only leverage a very small training set per subject (n = 48 over two sessions, which should be controlled for additionally). We fear that this would seriously hinder the potential of MVPA analyses, and generate results that would lack statistical robustness.

Reviewer #2 (Remarks to the Author):

The authors have successfully addressed my concerns and now I recommend publication of the manuscript.

We thank R2 for their recommendation for publication.

REVIEWERS' COMMENTS:

Reviewer #1 (Remarks to the Author):

Thank you for your thoughtful responses to my replies. I am satisfied with the current manuscript. Congrats on a nice paper.